# Separation of Convective and Stratiform Precipitation Using Polarimetric Radar Data with A Support Vector Machine Method

Yadong Wang[1], Lin Tang[2], Pao-Liang Chang[3], and Yu-Shuang Tang[3]

[1]Electrical and Computer Engineering Department, Southern Illinois University Edwardsville, Illinois, USA
[2]Cooperative Institute for Mesoscale Meteorological Studies, University of Oklahoma, NOAA/OAR/National Severe Storms Laboratory, Norman, Oklahoma, USA
[3]Central Weather Bureau, Taipei, Taiwan

**Correspondence:** Yadong Wang (yadwang@siue.edu)

**Abstract.** A precipitation separation approach using a support vector machine method was developed and tested on a C-band polarimetric weather radar located in Taiwan (RCMK). Different from some existing methods requiring a whole volume scan data, the proposed approach utilizes the polarimetric radar data from the lowest unblocked tilt to classify precipitation echoes into either stratiform or convective type. The inputs of radar reflectivity, differential reflectivity, and the separation index are integrated into the classification through a support vector machine algorithm. The feature vector and weight vector in the support vector machine were optimized using well-classified training data. The proposed approach was tested with multiple precipitation events including a widespread mixed stratiform and convective event, a tropical typhoon precipitation event, and a stratiform precipitation event. In the evaluation, the results from the multi-radar-multi-sensor (MRMS) precipitation classification algorithm were used as the ground truth. The performances from the proposed approach were further compared with the approach using the separation index only. It was found that the proposed method can accurately classify the convective and stratiform precipitation, and produce better results than using the separation index only.

## 1 Introduction

Convective and stratiform precipitations exhibit a significant difference in precipitation growth mechanisms and thermodynamic structures (e.g., Houghton, 1968; Houze, 1993, 1997). Generally, a convective precipitation is associated with strong but small areal vertical air motion ($> 5$ m s$^{-1}$) (Penide et al., 2013), and delivers a high rainfall rate ($R$) (Anagnostou, 2004). On the other hand, stratiform precipitation is associated with weak updrafts/downdrafts ($< 3$ m s$^{-1}$) and relatively low $R$. Classifying a precipitation into either convective or stratiform type not only promotes the understanding of cloud physics but also enhances the accuracy of quantitative precipitation estimation (QPE). For these purposes, numerous methods using ground in situ measurements or satellite observations were developed during the past four decades (e.g., Leary and Jr., 1979; Adler and Negri, 1988; Tokay and Short, 1996; Hong et al., 1999).

Ground-based weather radars, such as Weather Surveillance Radar, 1988, Doppler (WSR-88D), are currently used in all aspects of weather diagnosis and analysis. Precipitation classification algorithms using single- or dual-polarization radars were developed during the past three decades. For a single-polarization radar, developed algorithms mainly rely on radar reflectivity

($Z$) and its derived variables (e.g., Biggerstaff and Listemaa, 2000; Anagnostou, 2004; Yang et al., 2013; Powell et al., 2016).

For example, Steiner et al. (1995) (hereafter SHY95) proposed a separation approach that utilizes the texture features derived from the radar reflectivity field. In this approach, a grid point in the $Z$ field is identified as the convective center if its value is larger than 40 dBZ, or exceeds the average intensity taken over the surrounding background by specified thresholds. Those grid points surrounding the convective centers are classified as convective area, and far regions are classified as stratiform. Penide et al. (2013) found that SHY95 may misclassify those isolated points embedded within stratiform precipitation or associated

with low cloud-top height. Powell et al. (2016) modified the SHY95′s approach, and the new approach can identify shallow convection embedded within large stratiform regions. A neural network based convective-stratiform classification algorithm was developed by Anagnostou (2004). Six variables were used in this approach as inputs including storm height, reflectivity at 2 km elevation, the vertical gradient of reflectivity, the difference in height, the standard deviation of reflectivity, and the product of reflectivity and height. Similar variables were also used in a fuzzy logic based classification approach proposed by

Yang et al. (2013).

Although these listed classification algorithms have been developed and validated for years, a new robust algorithm is motivated for the following two reasons. The first is to utilize only the low tilt radar data for classification. According to the U.S. Radar Operations Center (ROC), the WSR-88D radars are currently operated without updating a complete volume during each volume scan, especially during precipitation events. New radar scanning schemes are designed to reorganize the updating order

for a high frequency in low elevations and a less frequency for high elevations. An alternative scanning scheme enables the WSR-88D radars are able to promptly capture the storm development for weather forecast and to obtain a more accurate precipitation estimation. These new schemes include the automated volume scan evaluation and termination (AVSET), supplemental adaptive intra-volume low-level scan (SAILS), the multiple elevation scan option for SAILS, and the mid-volume rescan of low-level elevations (MRLE). Under these new scanning schemes, the separation of stratiform/convective becomes a challenge

for those algorithms requiring a full volume scan of data. The second reason is to further explore the applications of the polarimetric variables. Polarimetric weather radars have been well applied in radar QPE, severe weather detection, hydrometeor classification, and microphysical retrievals (Ryzhkov and Zrnic, 2019; Zhang, 2016). Extra information about hydrometeors' size, shape, species, and orientation could be extracted through transmitting and receiving electromagnetic waves along the horizontal and vertical directions. Therefore, the polarimetric measurements may reveal more precipitation's microphysical

and dynamic properties. Inspired by these features, a C-band polarimetric radar precipitation separation approach was developed by Bringi et al. (2009) (hereafter BAL), which classifies the precipitation into stratiform, convective and transition regions based on retrieved drop size distribution (DSD) characteristics. However, it was found that strong stratiform echoes might have similar DSDs to weak convective echoes and lead to wrong classification results (Powell et al., 2016).

In this work, a novel precipitation separation algorithm using the separation index with other radar variables was developed

and tested on a C-band polarimetric radar located in Taiwan. This approach classifies precipitations into stratiform or convective type with a support vector machine (SVM) method. Different from some existing classification techniques that require a whole volume scan of radar data, this new approach uses the lowest unblocked tilt data in the separation. If the lowest tilt is partially or completely blocked, then the next adjacent unblocked tilt is used instead. The major advantage of this method is that it can

provide real-time classification results even if the radar is operated under AVSET, SAILS, and MRLE scanning schemes, where the low tilts are frequently scanned and updated. This paper is organized as follows: Section 2 introduces the proposed method including radar variables and processings, the SVM method, and the training process. The performance evaluation is shown in Section 3, and the discussion and summary are given in Section 4.

## 2    Precipitation Separation With a Support Vector Machine Method

In the current work, the SVM precipitation separation approach was developed and validated on a C-band polarimetric radar (RCMK) located at Makung, Taiwan (Figure 1). The Weather Wing of the Chinese Air Force deployed this radar and made the data available to the Central Weather Bureau (CWB) of Taiwan since 2009. Together with three single-polarization S-band WSR-88D (RCCG, RCKT, and RCHL) and one dual-polarization S-band radar (RCWF), these five radars provide real-time QPE products to CWB to support missions of flood monitoring and prediction, landslide forecasts and water resource management. Operating with a wavelength of 5.291 cm, RCMK performs volume scans of 10 tilts ($0.5°$, $1.4°$, $2.4°$, $3.4°$, $4.3°$, $6.0°$, $9.9°$, $14.6°$, $19.5°$, and $25°$) in every 5 minutes with the range resolution of 500 m and angular sampling of $1°$.

The Central Mountain Range (CMR) of Taiwan is also shown in Figure1, which poses a major challenge for radar based products. Radars located in complex terrain are prone to partial or total blockages, which cause data from the low elevation angles (EA) to be unavailable or problematic. Blockage maps of RCMK are illustrated in Figure 2. Since there are severe blockages at the $0.5°$ for RCMK, data from the $1.4°$ EA is used in the algorithm development.

### 2.1    Input polarimetric radar variables and preprocesses

Three measured or derived radar variables are proposed as inputs to the SVM approach: $Z$, differential reflectivity fields ($Z_{DR}$), and separation index ($i$). In most of precipitation classification approaches, $Z$ is used as one of the inputs because reflectivity from convective generally show higher values than from stratiform type. For example, a radar echo, with the reflectivity of 40 dBZ and above, is automatically classified as convective type in the approach developed by SHY95.

Differential reflectivity, which is highly related to raindrop's mass weighted mean diameter ($D_m$), is another good indicator of precipitation type. It was found the values of $D_m$ in stratiform and convective precipitation generally are within 1-1.9 mm and above 1.9 mm, respectively (Chang et al., 2009). Higher $Z_{DR}$ values are expected from convective than from stratiform precipitation. Therefore, the $Z_{DR}$ field is used as another input of the proposed approach.

For short wavelength radars such as C-band or X-band radars, the $Z$ and $Z_{DR}$ fields may be significantly attenuated when radar beam propagates through heavy precipitation regions. Both $Z$ and $Z_{DR}$ fields need to be corrected from attenuation before applied in the precipitation classification and QPE. Different attenuation correction methods were proposed using the differential phase ($\phi_{DP}$) measurement such as the linear $\phi_{DP}$ approach, the standard ZPHI method, and the iterative ZPHI method (e.g., Jameson, 1992; Carey et al., 2000; Testud et al., 2000; Park et al., 2005). Because of its simplicity and easy

implementation in a real-time system, the linear $\phi_{DP}$ method was applied in the current work.

$$Z(r) = Z'(r) + \alpha(\phi_{DP}(r) - \phi_{DP}(0)) \tag{1a}$$

$$Z_{DR}(r) = Z'_{DR}(r) + \beta(\phi_{DP}(r) - \phi_{DP}(0)) \tag{1b}$$

where $Z'(r)$ ($Z'_{DR}(r)$) is the observed reflectivity (differential reflectivity) at range $r$; $Z(r)$ ($Z_{DR}(r)$) is the corrected value; $\phi_{DP}(0)$ is the system value; $\phi_{DP}(r)$ is the smoothed (by FIR filter) differential phase at range $r$. The attenuation correction coefficients $\alpha$ and $\beta$ depend on DSD, drop size shape relations (DSR), and temperature. The typical range of $\alpha$ ($\beta$) is found 0.06~0.15 (0.01~0.03) dB deg$^{-1}$ for C-band radars (e.g., Carey et al., 2000; Vulpiani et al., 2012). Following the work from Wang et al. (2014), optimal coefficients $\alpha$ and $\beta$ in Taiwan are 0.088 dB deg$^{-1}$ and 0.02 dB deg$^{-1}$, respectively. The $Z$ and $Z_{DR}$ fields are further smoothed with a 3 (azimuthal) by 3 (range) moving window function after corrected from attenuation.

Other quality control issues, including calibration, reflectivity vertical profile, and ground clutter removal, were also considered in this work. Since this radar is used in the real-time quantitative precipitation estimation, the biases of $Z$ and $Z_{DR}$ should be within 1 dBZ, and 0.1 dB, respectively. The data quality of RCMK was examined through validating the QPE performance in different works (e.g., Wang et al., 2013, 2014). Therefore, the calibration bias of RCMK should be within the reasonable range. A vertical profile of reflectivity (VPR) correction is generally needed on the reflectivity field to reduce the measurement biases because of the melting layer (Zhang et al., 2011). Given the fact that 1.4° elevation angle is used within the maximum range of 150 km, and the melting layer is usually around 5 km in Taiwan, the radar data is well below the melting layer. In addition, considering the vertical profile of differential reflectivity is not well studied in the current stage, no vertical corrections are applied to fields of $Z$ and $Z_{DR}$. Ground clutter is typically associated with a low correlation coefficient ($\rho_{HV}$), the $\rho_{HV}$ threshold used in this work is 0.9, which can effectively remove those non-meteorological echoes such as ground clutter.

Using the separation index $i$ to identify convective from stratiform precipitation was initially proposed by BAL, where $i$ was calculated under a normalized gamma DSD assumption:

$$i = log_{10}(N_W^{est}) - log_{10}(N_W^{sep}) \tag{2}$$

$$log_{10}(N_W^{sep}) = -1.6D_0 + 6.3 \tag{3}$$

where $N_W^{est}$ is the estimated $N_W$ (normalized number concentration) from observed $Z$ and $Z_{DR}$, and is calculated as:

$$N_W^{est} = Z/0.056D_0^{7.319} \tag{4}$$

In Equation 4, $D_0$ is the median volume diameter and can be calculated as.

$$D_0 = 0.0203Z_{DR}^4 - 0.1488Z_{DR}^3 + 0.2209Z_{DR}^2 + 0.5571Z_{DR} + 0.801; \quad -0.5 \leq Z_{DR} < 1.25 \tag{5a}$$

$$= -0.0355Z_{DR}^3 - 0.3021Z_{DR}^2 + 1.0556Z_{DR} + 0.6844; \quad 1.25 \leq Z_{DR} < 5 \tag{5b}$$

The units of $Z_{DR}$, $Z$, $N_w$, and $D_0$ are dB, mm$^6$m$^{-3}$, mm$^{-1}$m$^{-3}$, and mm, respectively. The positive and negative values of index $i$ indicate convective and stratiform rain, respectively, and $|i| < 0.1$ indicates transition regions (Penide et al., 2013).

BAL pointed out that index $i$ worked well in most of the cases in their study; however, incorrect classification results are likely obtained for low $Z$ and high $Z_{DR}$ cases in some convective precipitations.

## 2.2   Drop size distribution and drop shape relation

It should be noted that the relations between $i$, $N_w$, and $D_0$ were derived using the DSD data collected in Darwin, Australia. Coefficients in Equations 2∼5 need be adjusted according to the radar frequency or/and DSD and DSR features from the
specific location (Thompson et al., 2015). In the current work, the separation index $i$ is directly derived using Equations 2∼5 without further adjustment. It was shown by Wang et al. (2013) that DSD and DSR features in Taiwan are very similar to those measured from Darwin, Australia. Similar $R(K_{DP})$ relationships were obtained using data collected from these two locations. Coefficients derived by BAL could be directly used in Taiwan without further modification. To verify this assumption, $N_w$ and $D_0$ were calculated using DSD data collected by four impact-type Joss-Waldvogel disdrometers (JWD)
located in Taiwan (Figure 1). The measurement range and temporal resolution of these JWDs are 0.359 mm ∼ 5.373 mm and 1 minute, respectively. A total of 4306-minute data from 2011∼2014 are used in $N_w$ and $D_0$ calculation following the approach described in Bringi et al. (2003). Similar to the work presented in BAL, the distribution of $i$ along median volume diameter $D_0$ is shown in Figure 2, where $(log_{10}N_w, D_0)$ pairs from stratiform and convective types are represented with gray circles and black stars, respectively. Although the relation described in Equation 3 can separate most stratiform from convective type, a
large number of points are still classified incorrectly. Therefore, the single separation index is not sufficient in the precipitation separation, and other variables such as $Z$ and $Z_{DR}$ may be used as supplements.

## 2.3   Support vector machines (SVM) method

### 2.3.1   Introduction of SVM

Machine learning algorithms based on meteorological radar data were well developed during the past two decades (e.g.,
Capozzi et al., 2018; T. et al., 2019; Yen et al., 2019). Support vector machine (SVM) can be viewed as a kernel-based machine learning approach, which nonlinearly maps the data from the low-dimension input space to a high-dimension feature space, and then linearly maps to a binary output space (Burges, 1998). Given a set of training samples, the SVM constructs an optimal hyperplane, which maximizes the margin of separation between positive and negative examples (Haykin, 2011). Specifically, given a set of training data $\{(X_i, y_i)\}_{i=1}^{N}$, the goal is to find the optimal weights vector $W$ and a bias $b$ such that

$$y_i(W^T X_i + b) \geq 1 \qquad i = 1, 2, ...., N \tag{6}$$

where $X_i \in \mathbb{R}^m$ is the input vector, $m$ is the variable dimension ($m = 3$ in this work), $N$ is the number of training samples, and $y_i$ is the output with the value of $+1$ or $-1$ that represents convective or stratiform, respectively. The particular data points $(X_i, y_i)$ are called support vector when Equation 6 is satisfied with the equality sign. The optimum weights vector $W$ and bias $b$ can be obtained through solving the Lagrangian function with the minimum cost function (Haykin, 2011).

Since the SVM can be viewed as a kernel machine, finding the optimal weight vector and bias in Equation 6 can be alternatively solved through the recursive least square estimations of:

$$\sum_{i=1}^{N_s} \alpha_i y_i k(X, X_i) = 0 \tag{7}$$

where $N_s$ is the number of support vectors, $\alpha_i$ is the Lagrange multipliers, and $k(X, X_i)$ is the Mercer kernel defined as:

$$k(X, X_i) = \Phi^T(X_i)\Phi(X) = exp\left(-\frac{1}{2\sigma^2}||X - X_i||^2\right) \tag{8}$$

With the solved $\{\alpha_i\}_{i=1}^{N}$, the SVM calculate the classification results with new input data $Z \in \mathbb{R}^m$ as:

$$f(Z) = sign\left[\sum_{i=1}^{N_s} \alpha_i y_i \Phi^T(X_i)\Phi(Z)\right] \tag{9}$$

When $f(Z) = 1$, the output is classified as convective, otherwise is classified as stratiform.

### 2.3.2 Training of the SVM

In the SVM approach, the weight vector and bias in Equation 6 need to be optimized through a recursive least square estimation using training data. Since the training data play a critical role in the SVM approach, $Z$, $Z_{DR}$ and $i$ from convective and stratiform precipitation events were carefully examined through three steps. Firstly, the training data was checked following general classification principles. For example, training data from convective precipitation is generally associated with relatively strong reflectivity and high vertically integrated liquid (VIL). On the other hand, stratiform precipitations are generally associated with a prominent bright band signature. The melting hydrometeors within the bright band increase backscatter during stratiform rainfall, which can significantly enhance radar reflectivity. The bright band feature is one of the obvious indicators of stratiform precipitation. Bright band signature normally can be observed from relatively high EAs (such as above 9.9°). From low EAs, because of the combination of radar beam broadening and low slant angle, the bright band feature spreads into more gates and becomes not apparent. Therefore, in this work, the bright band feature from high elevation angles is only used in training data selection but not used as one of the inputs. Secondly, the precipitation type is verified by ground observation, such as ground severe storm reports. Thirdly, the precipitation type is confirmed by the Multi-Radar-Multi-Sensor (MRMS) precipitation classification algorithm implemented in Taiwan (Zhang et al., 2011, 2016). In this MRMS classification approach, a three-dimensional radar reflectivity field was mosaicked from 4 S-band single-polarization radars (Figure 1), The composite reflectivity (CREF) and other measurements such as temperature and moisture fields were then used in the surface precipitation classification (Zhang et al., 2016). Based on the classification results, MRMS chooses different $R(Z)$ relations in the rainfall rate estimation. The performance of MRMS has been thoroughly evaluated for years for the quantitative precipitation estimation, flash flood monitoring, severe weather, and aviation weather surveillance (e.g., Gourley et al., 2016; Smith et al., 2016). The products are used as the benchmark and/or ground truth in many studies (e.g., Grecu et al., 2016; Skofronick-Jackson and Coauthors, 2017). It should be noted that, on the other hand, the MRMS also shows limitations since it only uses

single-polarization variables to determine the precipitation type. At the current stage, the MRMS precipitation classification is considered as the appropriate benchmark in the training and validation of the proposed algorithm. Moreover, since the MRMS classification is a mosaicked product derived from 4 S-band radars, it can be viewed as an independent reference.

Convective type training data is mainly from a strong convective precipitation event on 23 July 2014. This thunderstorm, classified as convective precipitation by MRMS, was associated with strong updrafts/downdrafts and caused an aircraft crash on the airport of Makung at 1106 UTC. The squall line features can be clearly identified from this storm. Radar data collected from 1030 to 1130 UTC were used as the convective type training data. The training data selection follows the criteria of $Z >$ 20 dBZ, and correlation coefficient ($\rho_{HV}$) > 0.98 (Kumjian, 2013). The stratiform type data are from a mixed stratiform and convective precipitation event on 30 August 2011, and only those data identified as a stratiform type by MRMS are used in training. A total of 17281 sets of data (15144 sets of stratiform, and 2137 sets of convective) are used in the training process. In this work, one data set is defined as the variables from a gate in terms of range and azimuthal angle. Be more specific, a set of training data means a vector of $[Z(a,r)\ Z_{DR}(a,r)\ i(a,r)\ d(a,r)]$, where $a$ and $r$ indicate azimuthal angle and range, respectively. The variable $d$ is the ground truth (with 1 and -1 represents convective and stratiform), which is as the desired response in the training process. The number of support vectors is selected as 1000 in the current work, and the training process is considered as completed when the root-mean-square error reaches a stable value. In the SVM approach, the original three-dimension input space nonlinearly maps to a 1000-dimension feature space, and then linearly maps to a binary output space (Burges, 1998). The higher dimension feature space potentially captures more input variables features with higher computation cost. Generally, after the number of support vector reaches some number, the enhancement in the SVM's performance approach becomes slight. There is a balance between accuracy and computation. In the current work, the number of support vectors was tested with a value of 500, 750, 1000, 2000, and 5000. The testing of 1000 support vectors can produce less than 5% error with reasonable computation time. As the prototype algorithm, the number of support vectors is selected as 1000 in the current work.

## 3  Performance Evaluation

### 3.1  Description of the experiments

The performance of the proposed approach was validated with three precipitation events from 2009 to 2011. These three precipitation events include one stratiform precipitation, one intense tropical precipitation, and one mixed convective and stratiform precipitation. Two experiments based on the BAL approach with different thresholds (i.e., BAL$^0$ and BAL$^{-0.5}$) were also carried out in the evaluation. In these two experiments, the separation index $i$ from each radar gate is first calculated using Equations 2~5, and thresholds of $T_0 = 0$ and -0.5 are then used to separate convective type from stratiform type. A gate is classified as convective if $i$ is larger than $T_0$, and as stratiform otherwise. This work aims to develop a complementary method using separation index $i$ and other variables to separate convective from stratiform type. The proposed SVM and BAL methods can classify the precipitation using the lowest tilt radar data only, which is suitable for fast scanning and quickly

updated purposes. Other classification approaches as introduced in section 1 were not examined in the current work, because they require the data from multiple elevation angles.

In the evaluation, three statistical scores of probability of detection (POD), false alarm rate (FAR), and critical success index (CSI) are first used, and MRMS classification results are used as the "ground truth" in the calculation.

$$POD = \frac{hit}{hit + miss} \tag{10}$$

$$FAR = \frac{false}{hit + false} \tag{11}$$

$$CSI = \frac{hit}{hit + false + miss} \tag{12}$$

where "hit," "false," and "miss" are defined as a radar gate classified as convective type by MRMS and the evaluated approach simultaneously, by the evaluated approach only, and by MRMS only, respectively. Although these scores are well used in statistical analysis, two factors make it necessary to introduce one more criterion in the evaluation. First, MRMS results are derived using the mosaicked field from four S-band single-polarization radars, and the classification results are produced every 10 minutes. On the other hand, $BAL^0$, $BAL^{-0.5}$, and SVM generate classification results whenever RCMK completes a whole scan. The time difference between results from RCMK ($BAL^0$, $BAL^{-0.5}$, and SVM) and MRMS could be as large as 5 minutes. Second, a convective storm's size, intensity, and cells locations could change significantly during a short period. Therefore, these three pixel-to-pixel based evaluation scores cannot really reflect the performance of the proposed approach. As a supplement, a whole coverage convective ratio ($R^{CS}$) is introduced in the current work:

$$R^{CS} = \frac{N^{con}}{N^{con} + N^{str}} \times 100\% \tag{13}$$

Where $N^{con}$ and $N^{str}$ are the total pixel numbers of convective and stratiform types within the coverage, respectively. Together with CSI, POD and FAR, these four scores are used in the performance qualitative validation. The evaluation results are shown in the following sections.

## 3.2 Experiment results

### 3.2.1 Widespread mixed stratiform and convective precipitations

The performance of the proposed approach was first validated with one widespread stratiform and convective mixed precipitation event on 30 August 2011, and 24-hour data (0000 UTC~2400 UTC) were used in the evaluation. Classification results from the proposed SVM were calculated with the trained weight vector and biases, and results from the BAL approach ($BAL^0$ and $BAL^{-0.5}$) were also calculated for the comparison purpose. It should be noted that the threshold of -0.5 is much lower than the value suggested by BAL, and $BAL^{-0.5}$ may classify more precipitations as convective type.

The time series of $R^{CS}$ (A), CSI (B), FOD (C), and FAR (D) are calculated using Equations 10∼13 and shown in Figure 4, where results from MRMS, SVM, BAL$^0$, and BAL$^{-0.5}$ are presented by black, red, blue, and green lines, respectively. When the MRMS results are applied as the ground truth, BAL$^0$ obviously classifies more precipitation as stratiform type during this 24-hour period. The time series of $R^{CS}$ from BAL$^0$ are much lower than other three approaches. BAL$^{-0.5}$ classifies more pixels as convective than BAL$^0$ as expected, and the $R^{CS}$ scores are much higher than BAL$^0$. The proposed SVM shows the

most similar $R^{CS}$ scores to MRMS comparing to BAL approaches. Since the BAL$^{-0.5}$ uses a very low threshold, it classifies more pixels as convective type, and the obtained $R^{CS}$ scores are higher than MRMS. In term of CSI, POD, and FAR, SVM and BAL$^{-0.5}$ show similar results, but BAL$^0$ show apparently worse performance.

To better understand the performance of each approach, the classification results and radar variables ($Z$, $Z_{DR}$, and $i$) from two distinct moments are examined and shown in Figures 5∼7. Figure 5 shows the classification results from 0303 UTC 30

August 2011, where BAL$^0$, BAL$^{-0.5}$, SVM and MRMS are shown in the panel 'A', 'B', 'C', and 'D', respectively. The time stamp for the MRMS result is 0300 UTC, and about 3 minutes earlier than the other three approaches. These three input variables of SVM at 0303 UTC are shown in Figure 6, where $Z$, $Z_{DR}$, and $i$ are presented in panel 'A', 'B', and 'C'. From Figures 4 and 5, it could be found that the $R^{CS}$ from MRMS, SVM, and BAL$^{-0.5}$ show similar values, but $R^{CS}$ from BAL$^0$ is distinctly low. Within the red circle of Figure 6, the averages of $Z$ and $Z_{DR}$ both show relatively large values ($Z > 36$ dBZ

and $Z_{DR} > 0.75$ dB), this is a clear indication of convective type precipitation. Both SVM and BAL$^{-0.5}$ classify most of the area within the red circle as convective, and this result is consistent with the MRMS result. Since the separation indexes within the black circle are below or slightly higher than 0, most of the area is classified as stratiform type by BAL$^0$. For this moment, threshold $-0.5$ shows better performance than 0.

Figure 7 shows another example of classification results from 0650 UTC. At this moment, although SVM and BAL$^{-0.5}$

produce similar CSI (0.30 v.s. 0.25) and POD (0.48 v.s. 0.52), the $R^{CS}$ from BAL$^{-0.5}$ (32%) is much higher than $R^{CS}$ from MRMS (17%) and SVM (13%). These scores could also be found in Figure 4. In Figure 7, It could be found from that the MRMS, SVM, and BAL$^{-0.5}$ show similar classification results between the azimuthal angle of $180°$ and $270°$. However, BAL$^{-0.5}$ misclassifies gates between $90°$ and $180°$ as convective type, which produces such high $R^{CS}$. On the other hand, MRMS and SVM show similar classification results in this region.

### 3.2.2   Tropical convective

Typhoon Morakot (6∼10 August 2009) brought significant rainfall to Taiwan. Over 700 people were reported dead in the storm, and the property loss was more than 3.3 billion USD. For most of the time during its landfall in Taiwan, the precipitation was classified as a mixture of tropical convective and tropical stratiform types. The performances of SVM, BAL$^0$, and BAL$^{-0.5}$ were validated using 96-hour data from 6 to 9 August 2009, where the results from 10 August 2009 were not included in the

evaluation because no significant precipitation was observed from that day. The time series plots of $R^{CS}$ (A), CSI (B), POD (C) and FAR (D) are shown in Figure 8. It could be found that scores of $R^{CS}$, CSI, and POD from the BAL based approaches is evidently lower than the results from SVM and MRMS, and the latter two show similar performance during these four days.

Classification results from $BAL^0$, $BAL^{-0.5}$, SVM (0402 UTC), and MRMS (0400 UTC) from 9 August 2009 are shown in Figure 9A, 9B, 9C, and 9D, respectively. The classification results in those regions, highlighted with two circles, are convective (SVM and MRMS) and stratiform ($BAL^0$ and $BAL^{-0.5}$). Figure 10 includes the reflectivity (A), differential reflectivity (B), and separation index (C) from 0402 UTC, where Figure 10D is the zoom-in reflectivity field inside the red rectangular box (A) for more details. It was found that the heavy precipitation band is on the top of RCMK (Figure 10D), and this may cause significant attenuation on $Z$ and $Z_{DR}$ fields. Although both $Z$ and $Z_{DR}$ fields were corrected using Equation 1, deficient or over compensations on $Z$ and $Z_{DR}$ fields lead to increased uncertainty on the separation index. It may be the primary reason causing the small values of the separation index. Other reasons such as wet radome may also contribute to the $Z$ and $Z_{DR}$ issues. In Figure 10C, the separation index $i$ are equal or less than -0.5 in the circled areas, and the BAL based approaches classify these regions as stratiform. On the other hand, these regions clearly show the convective precipitation features in the fields of $Z$ (10A) and $Z_{DR}$ (10D).

### 3.2.3 Stratiform precipitation event

The performances of $BAL^0$, $BAL^{-0.5}$, and SVM approaches were also evaluated with a widespread stratiform precipitation event on 26 March 2011. There were no convective type precipitations identified by MRMS, and all these three approaches showed consistent classification results with the MRMS result during an 8-hour period evaluation.

### 3.3 Sensitivity test

The performances of $BAL^0$, $BAL^{-0.5}$ and proposed SVM were further validated respecting to the $Z_{DR}$ bias. First, the impact of $Z_{DR}$ bias on $i$ is investigated through a simple simulation. In the simulation, the separation index $i$ is calculated using Equations 2∼5 with four distinct $Z$ values: 10 dBZ, 20 dBZ, 30 dBZ, and 40 dBZ. For each $Z$ value, $Z_{DR}$ changes from -0.5 dB to 2 dB to simulate the $Z_{DR}$ bias. The obtained $i$ results are shown in Figure 11, and the symbol of triangle, diamond, cross, and pentagram indicates the result from 10 dBZ, 20 dBZ, 30 dBZ, and 40 dBZ, respectively. It could be found that for each $Z$, the calculated $i$ drops when $Z_{DR}$ increases. Moreover, a larger $Z$ produces a larger $i$ for the same $Z_{DR}$ value. As introduced in Section 2.1, the precipitation may be classified as stratiform when $i$ is less than 0. Therefore, positive $Z_{DR}$ calibration bias may result in misclassifying more precipitation as stratiform type.

The impact of the $Z_{DR}$ calibration bias on the performance of $BAL^0$, $BAL^{-0.5}$, and SVM was investigated using precipitation events from 30 August 2011. In this study, the $Z_{DR}$ filed was first corrected from attenuation, and a $\Delta Z_{DR}$ was then manually added on the corrected $Z_{DR}$ field as the artificial bias. The $\Delta Z_{DR}$ was set as: -0.2 dB, -0.1 dB, 0 dB, 0.1 dB, and 0.2 dB, respectively. The biased $Z_{DR}$ was calculated as $Z_{DR}^b = Z_{DR} + \Delta Z_{DR}$. The separation index $i$ was calculated using $Z_{DR}^b$ through Equations 2∼5, and classification results from $BAL^0$ and $BAL^{-0.5}$ were then calculated. The same trained weights and bias vector described in Section 2.3.2 were used in the SVM approach. Following the procedure described in Section 3.2, scores of $R^{CS}$ (A), CSI (B), POD (C) and FAR (D) are calculated and shown in Figure 12. It should be noted that these scores are the 24-hour averaged values. It could be found that when the $\Delta Z_{DR}$ changes from -0.2 dB to 0.2 dB, the $R^{CS}$ from both $BAL^0$ and $BAL^{-0.5}$ approaches decrease. This indicates that $BAL^0$ and $BAL^{-0.5}$ classify more precipitation as stratiform,

and this results is consistent with the simulation. Both CSI and POD from $BAL^0$ and $BAL^{-0.5}$ show degradations with the increase of $\Delta Z_{DR}$. On the other hand, the proposed SVM shows slightly better performances when $\Delta Z_{DR}$ changes from negative to positive. Both CSI and POD increase when $\Delta Z_{DR}$ increases, and the $R^{CS}$ also has the similar trend. One possible reason is that convective type precipitation is normally associated with larger $Z_{DR}$. As a result, positive $\Delta Z_{DR}$ classify more precipitation as convective type. Similar results were also obtained from the case of 6~10 August 2009.

## 4    Conclusions

A novel precipitation classification approach using a support vector machine approach was developed and tested on a C-band polarimetric radar located in Taiwan. Different from other classification algorithms that use whole volume scan data, the proposed method only utilizes the data from the lowest unblocked tilt to separate precipitation into convective or stratiform type. It can be applied to new scanning schemes with more frequent scans at the lowest tilts and lack of information from a higher tilt, such as AVSET, SAILS, MRLE, and etc. Three radar variables of reflectivity, differential reflectivity, and the separation index derived by Bringi et al. (2009) are utilized in the new proposed approach, where both reflectivity and differential reflectivity need be corrected from attenuation and differential attenuation. Although the separation index alone can be used in the precipitation classification, there may be two potential limitations: thresholds and the biases on reflectivity and/or differential reflectivity. Although the threshold "0" was suggested to be used in separating convective type from stratiform type, it was found that a single threshold may not sufficient for all cases. Other thresholds (such as "$-0.5$" used in the current work), sometimes can produce better results than "0". The biases may come from mis-calibration, attenuation, wet radome, blockage. Although both reflectivity and differential reflectivity should be corrected from attenuation before used in the separation index calculation, the correction biases on either filed may cause large uncertainty in the derived separation index and further lead to a wrong classification. Other factors also may have impacts on the separation index. This work attempts to propose a complementary method to enhance the performance of using the separation index only. The proposed approach integrates input variables with a support vector machine method. The weighs and bias vectors used in the support vector machine were trained with typical stratiform and convective precipitation events. It should be noted that the proposed approach has a flexible framework, and some other variables can be easily included. With newly added variables, the weighting and bias vectors need to be retrained. The proposed approach was tested with multiple cases. Its performance was found similar to a well-developed approach, MRMS, which utilizes multiple tilts radar data in the classification. It should be noted that the time difference between RCMK (i.e., $BAL^0$, $BAL^{-0.5}$, and SVM) and MRMS could be as large as 5 minutes. Therefore, the pixel-to-pixel evaluation criteria of the critical success index (CSI), probability of detection (POD) and false alarm rate (FAR) may not really reflect their performances. Although a new variable of $R^{CS}$ is used in the performance evaluation, this should be treated as qualitative evaluation.

There are some issues that need to be noticed before applying this approach into operation. First, this approach is developed for fast scanning and fast update purpose, therefore, data from the lowest unblocked tilt is used as the input. However, if the radar is located in a complex orography area, radar beam could be partially or completely blocked at some regions. A possible

solution for such scenario is using data from different scanning tilts to form a hybrid scan, and the hybrid scan is then used as

the input. Radar scanning tilts used in the hybrid scanning are determined by the radar scanning geometry. Given the factor that precipitation's microphysics (such as drop size distribution) from different altitudes may be significantly different, therefore, the performance of proposed approach may be worse than expected. Second, the performance of the proposed approach depends highly on the training data, which should be selected very careful. Third, coefficients in the separation index calculation depend on the local drop size distribution and drop shape relation features. Therefore, new relations need to be derived for the optimal

results. Moreover, the separation index only validates at liquid phase precipitation. For ice phase precipitation, such as mixed hail and rain, its performance is not well studied. Other hydrometeor classification schemes could be used for such scenario. Fourth, this work only presents a prototype algorithm. Given the flexible framework, other variables (such as differential phase) could be easily integrated into this algorithm, and the performance could be further enhanced.

*Code and data availability.*

The datasets and source code used in this study are available from the corresponding author upon request (yadwang@siue.edu).

*Author contributions.*

The algorithm was originally developed by Dr. Y. Wang. Dr. L. Tang processed the radar data including generate results from MRMS. Dr. P.-L. Chang and Miss Y.-S. Tang provided and processed radar data from CWB, they were further involved in algorithm discussion and article writing.

*Competing interests.*

The authors declare that there is no conflict of interest.

*Acknowledgements.* The authors thank the radar engineers from CWB help us collecting and processing the radar data.

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

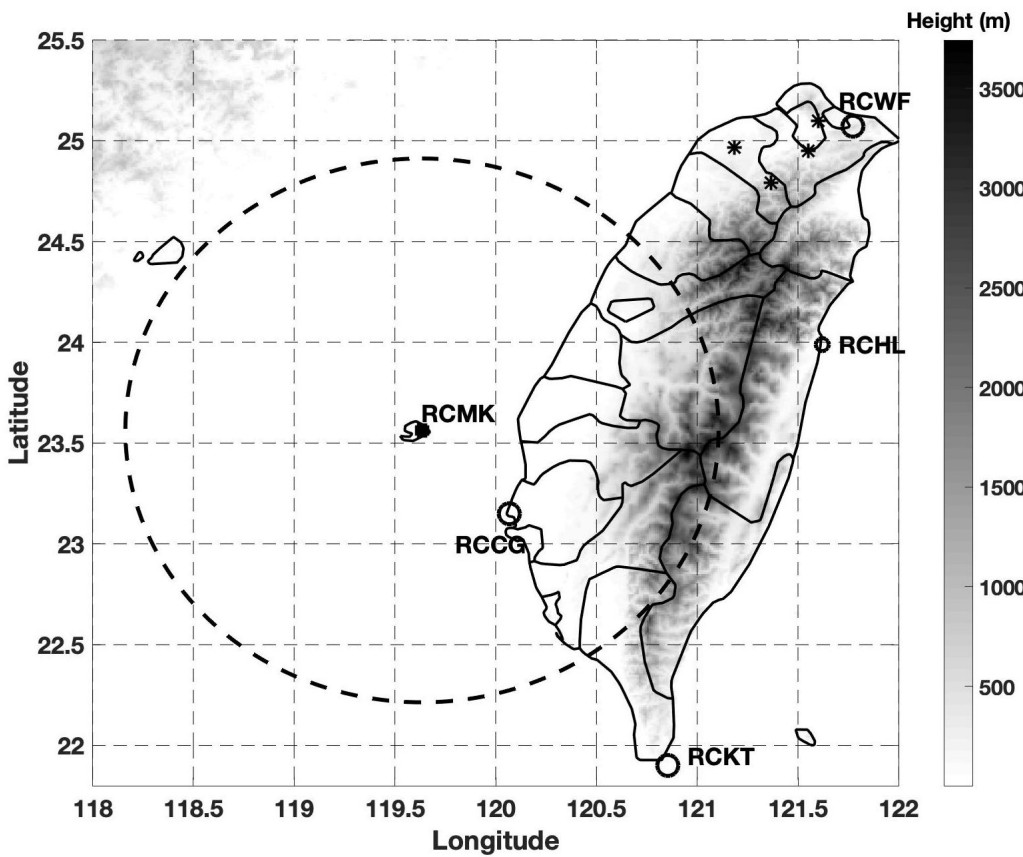

**Figure 1.** The terrain of Taiwan, the location of a C-band polarimetric radar RCMK (marked with a black square), JWDs (marked with black stars), and four S-band single-polarization radar RCCG, RCKT, RCHL, and RCWF (marked with black circles). The continuous grey-scale terrain map shows the central mountain range of Taiwan.

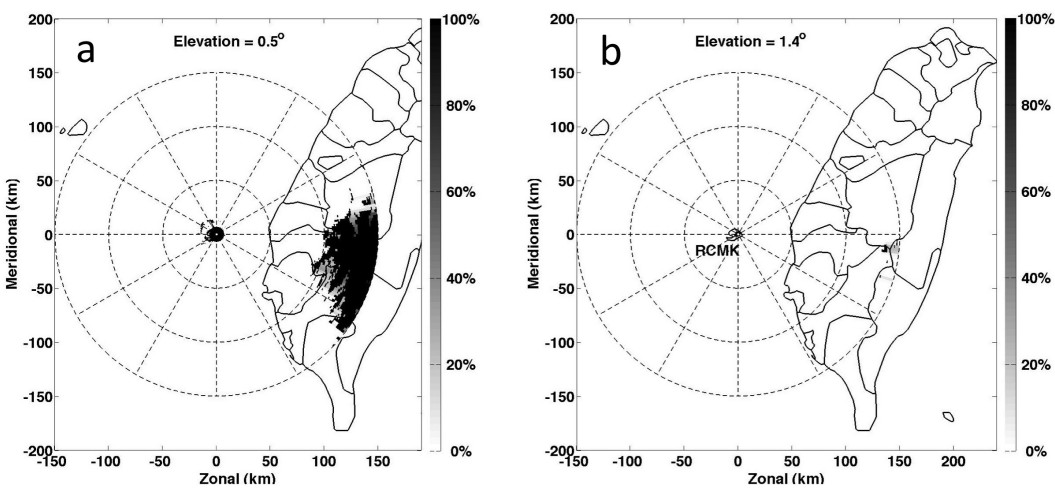

**Figure 2.** Blockage maps of RCMK from the first 2 EAs (0.5° and 1.4°). The grey scale indicates the blockage percentages.

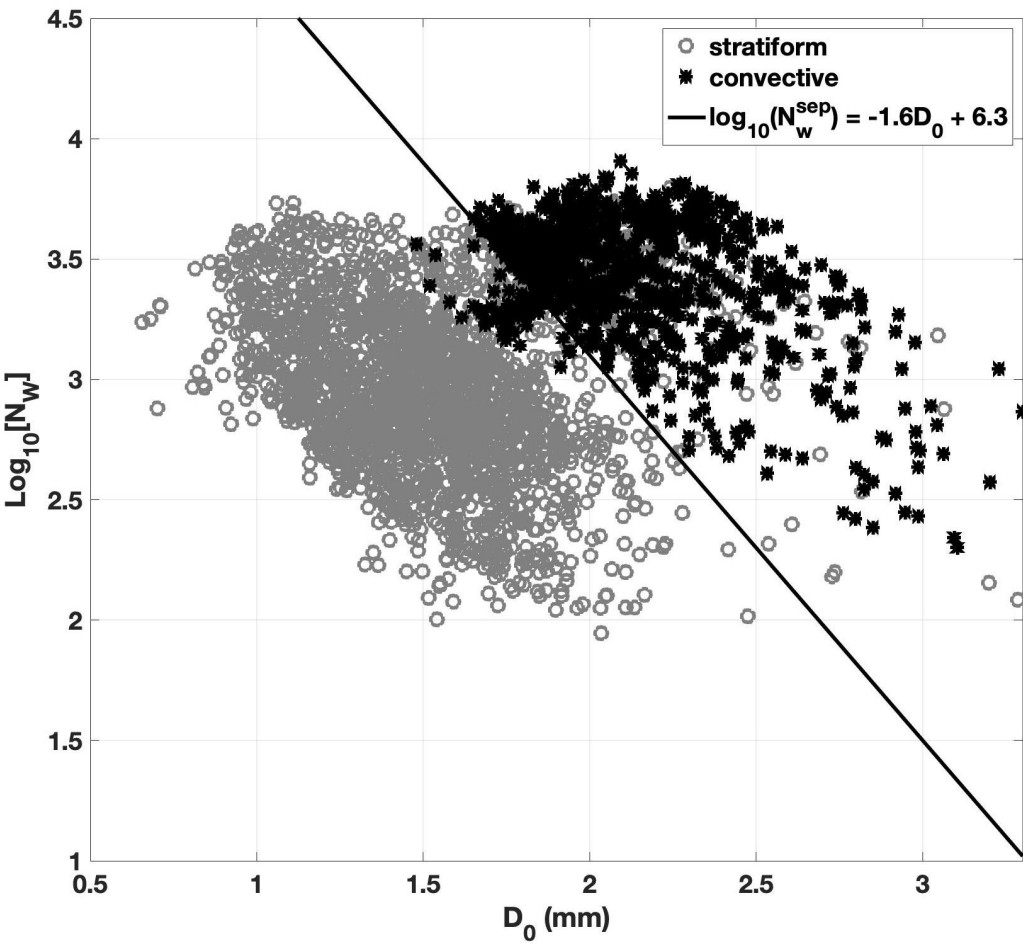

**Figure 3.** The distribution of $log10(N_w)$ vs $D_0$. The DSD data from stratiform and convective precipitations are presented with gray circles and black stars, and the separator line is shown with a solid line.

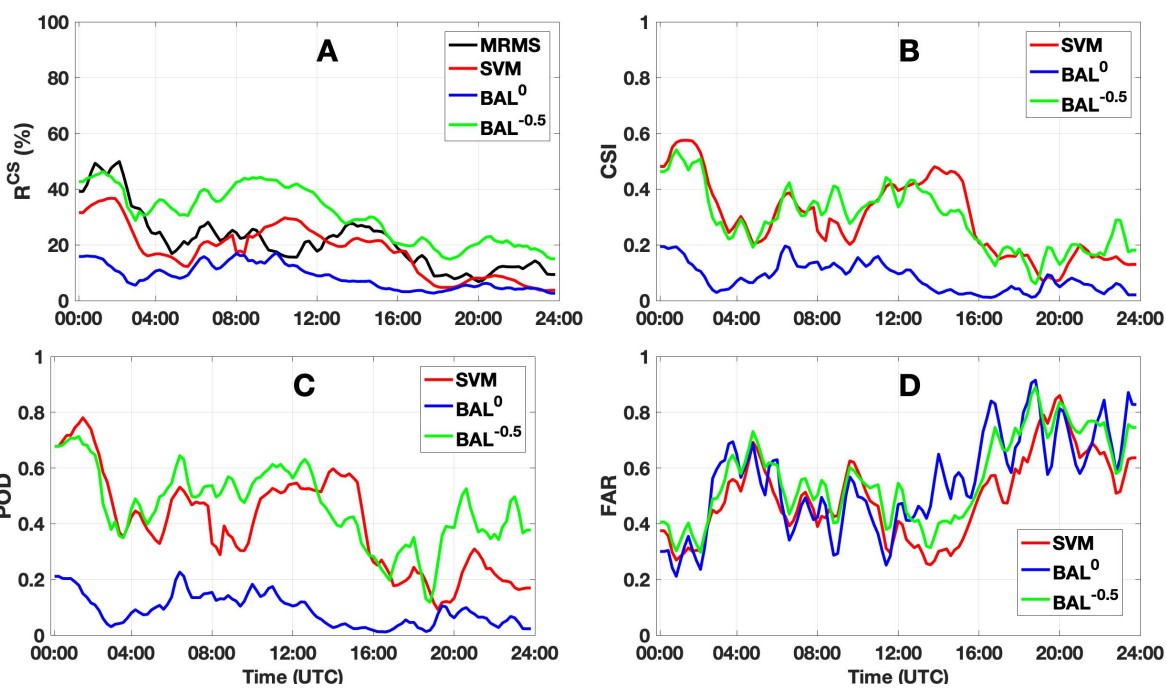

**Figure 4.** The time series plot of $R^{CS}$(A), CSI(B), POD(C), and FAR(D) from 30 August 2011. 24-hours data 0000 UTC  2400 UTC are used in each case. The results from BAL with threshold $T_0$ = -0.5, BAL with threshold $T_0$ = 0, SVM, MRMS, are presented by green, blue, red and black lines, respectively.

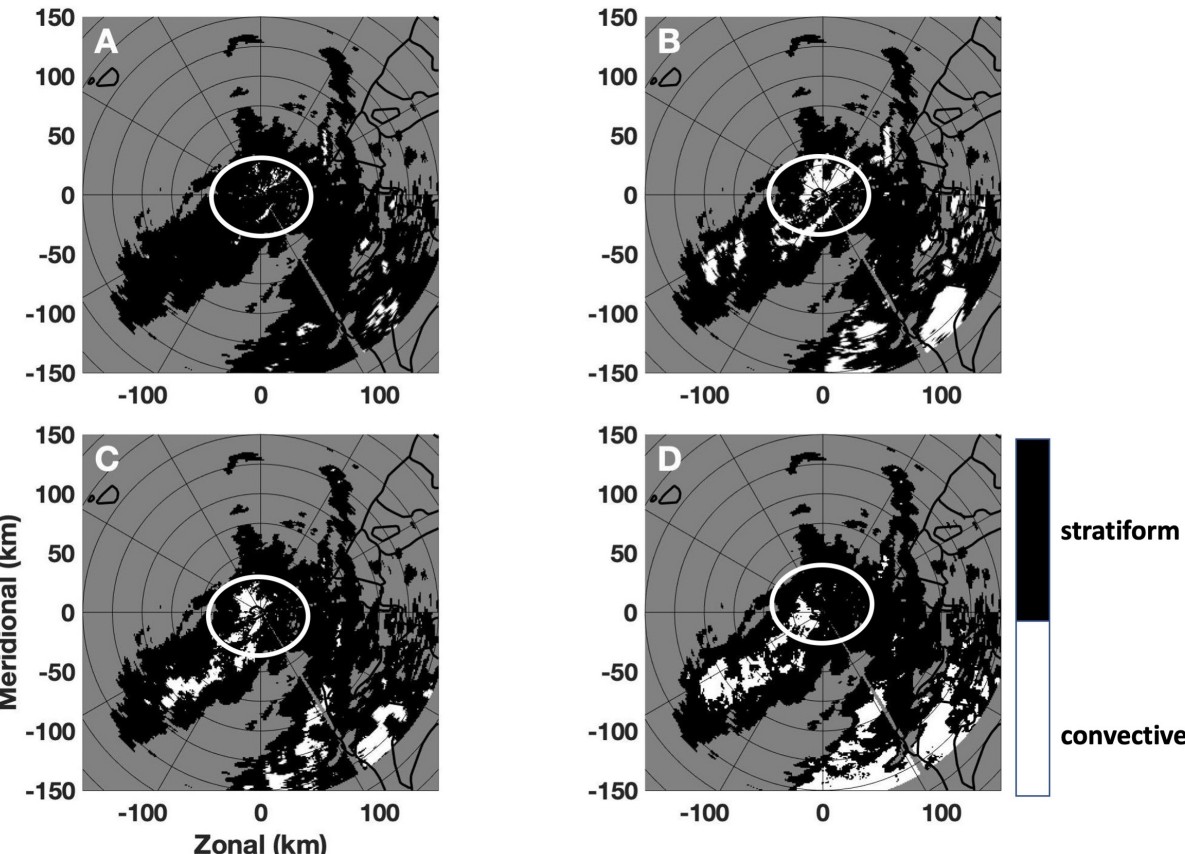

**Figure 5.** The classification results from $BAL^0$(A), $BAL^{-0.5}$(B), SVM(C) and MRMS(D). The time stamp for $BAL^0$, $BAL^{-0.5}$, and SVM is 0303 UTC 30 August 2011, and time stamp for MRMS is 0300 UTC 30 August 2011. The region inside the white circle is used in the analysis.

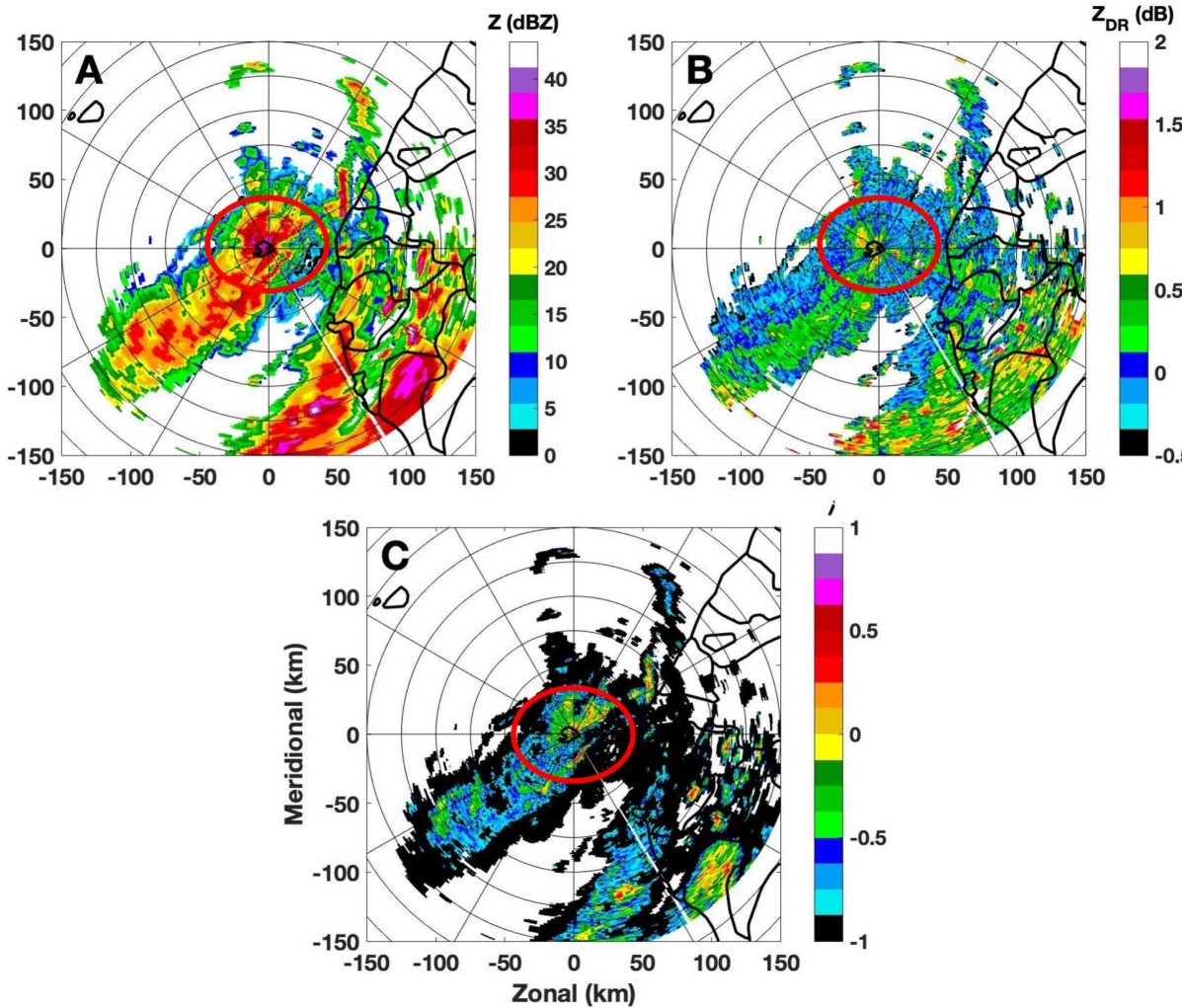

**Figure 6.** Radar variables of reflectivity (A), differential reflectivity(B), and separation index(C). The radar data was collected by RCMK at 0303 UTC 30 August 2011. The region inside the red circle is used in the analysis.

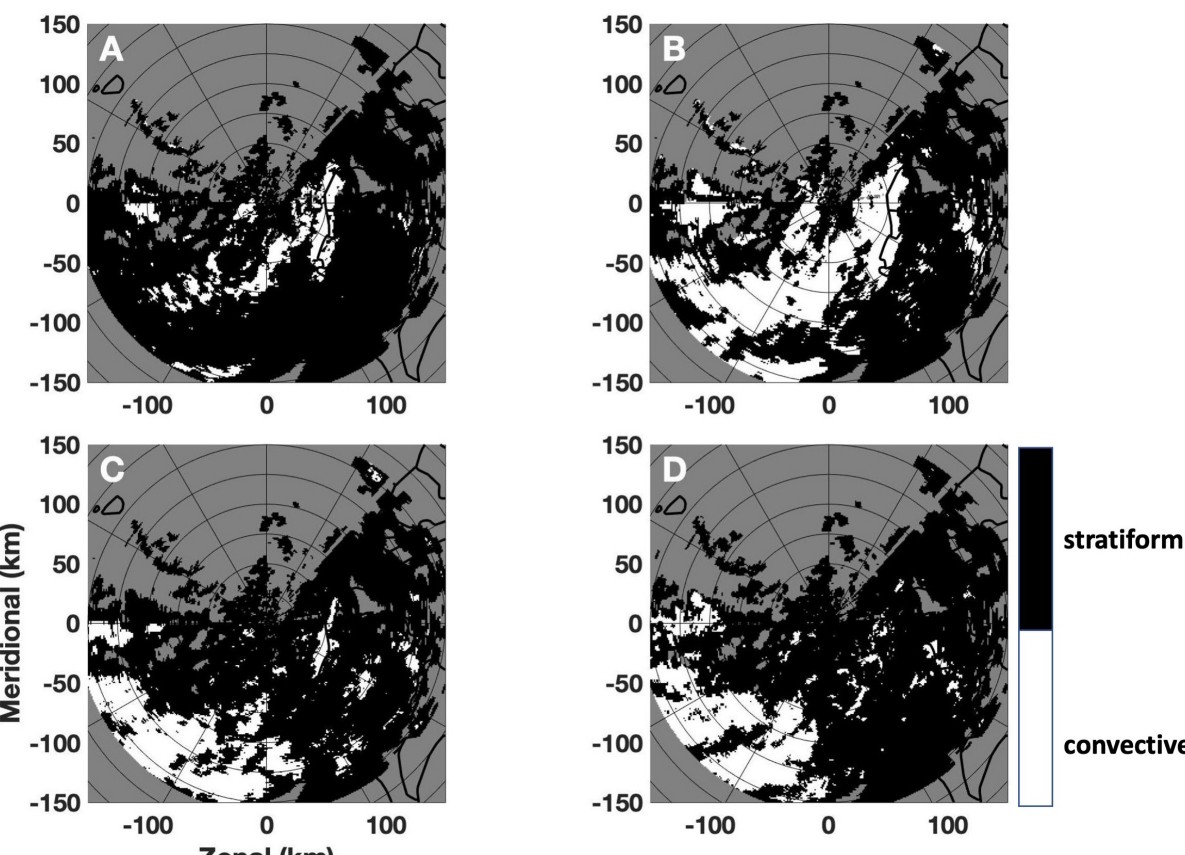

**Figure 7.** Similar to Figure 5, results are from 0650 UTC.

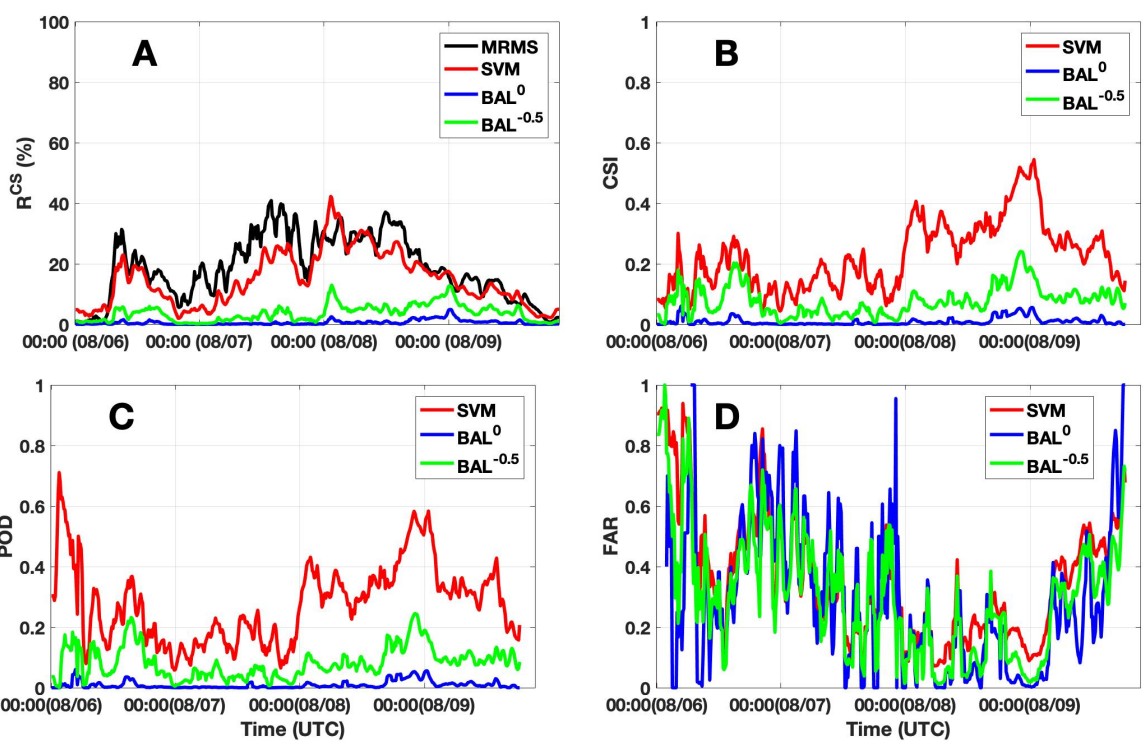

**Figure 8.** The time series plot of $R^{CS}$(A), CSI(B), POD(C), and FAR(D) from 06~09 August 2009. 96-hours data are used in each case. The results from BAL with threshold $T_0$ = -0.5, BAL with threshold $T_0$ = 0, SVM, and MRMS are indicated by green, blue, red and black lines, respectively.

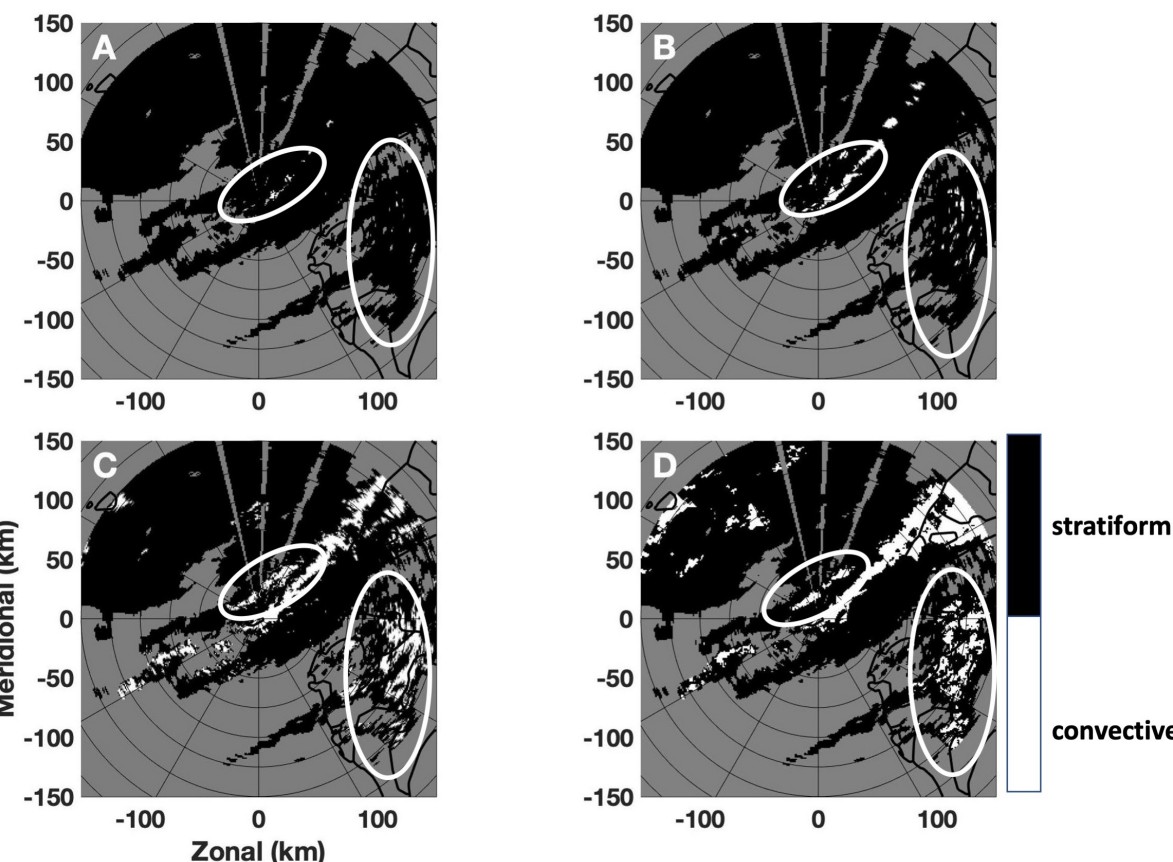

**Figure 9.** The classification results from $BAL^0$(A), $BAL^{-0.5}$(B), SVM(C), and MRMS(D). The time stamp for $BAL^0$, $BAL^{-0.5}$, and SVM is 0402 UTC 9 August 2009, and time stamp for MRMS is 0400 UTC 9 August 2009.

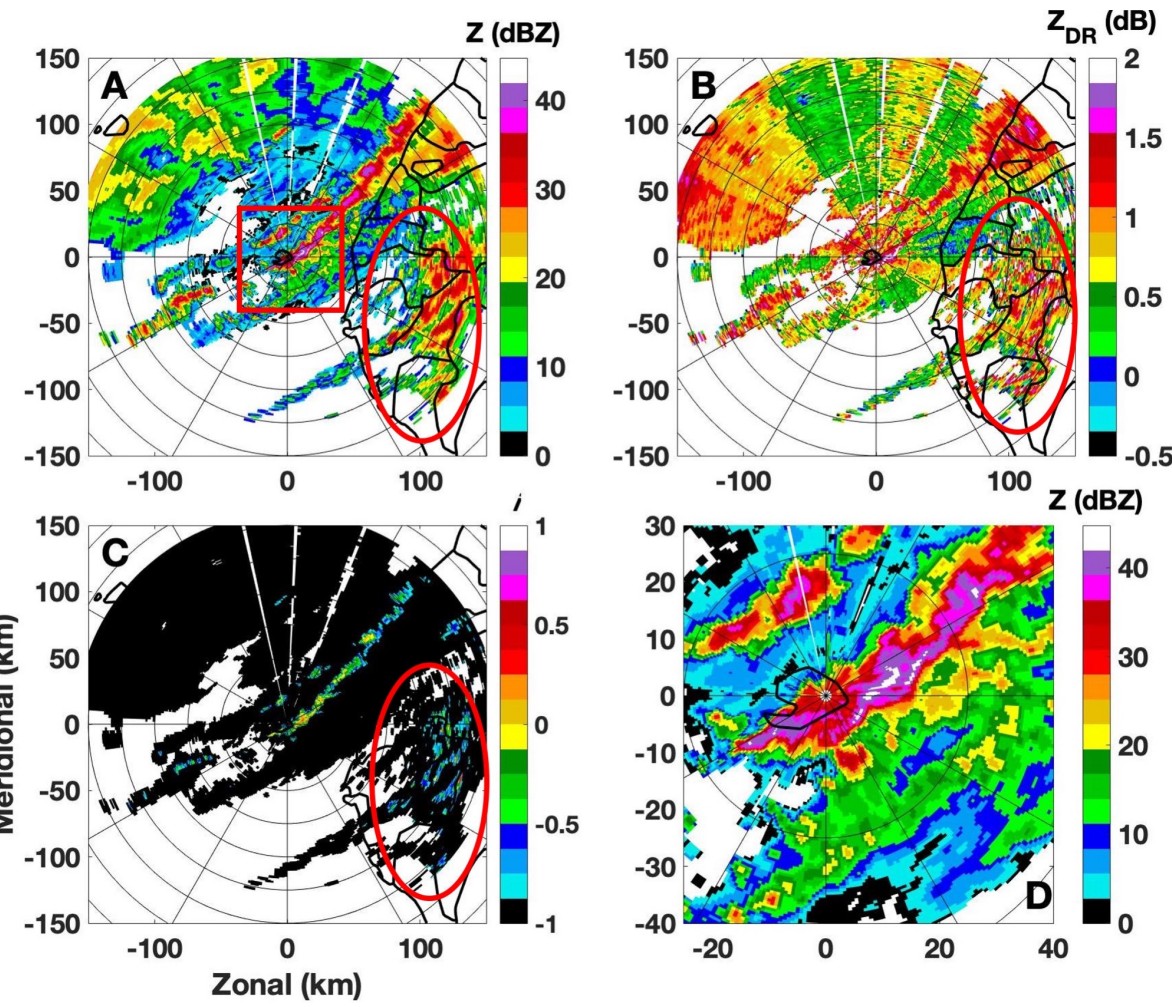

**Figure 10.** Radar variables of reflectivity(A), differential reflectivity(B), separation index(C), and reflectivity within the red rectangular box in A(D). The radar data was collected by RCMK at 0402 UTC 9 August 2009.

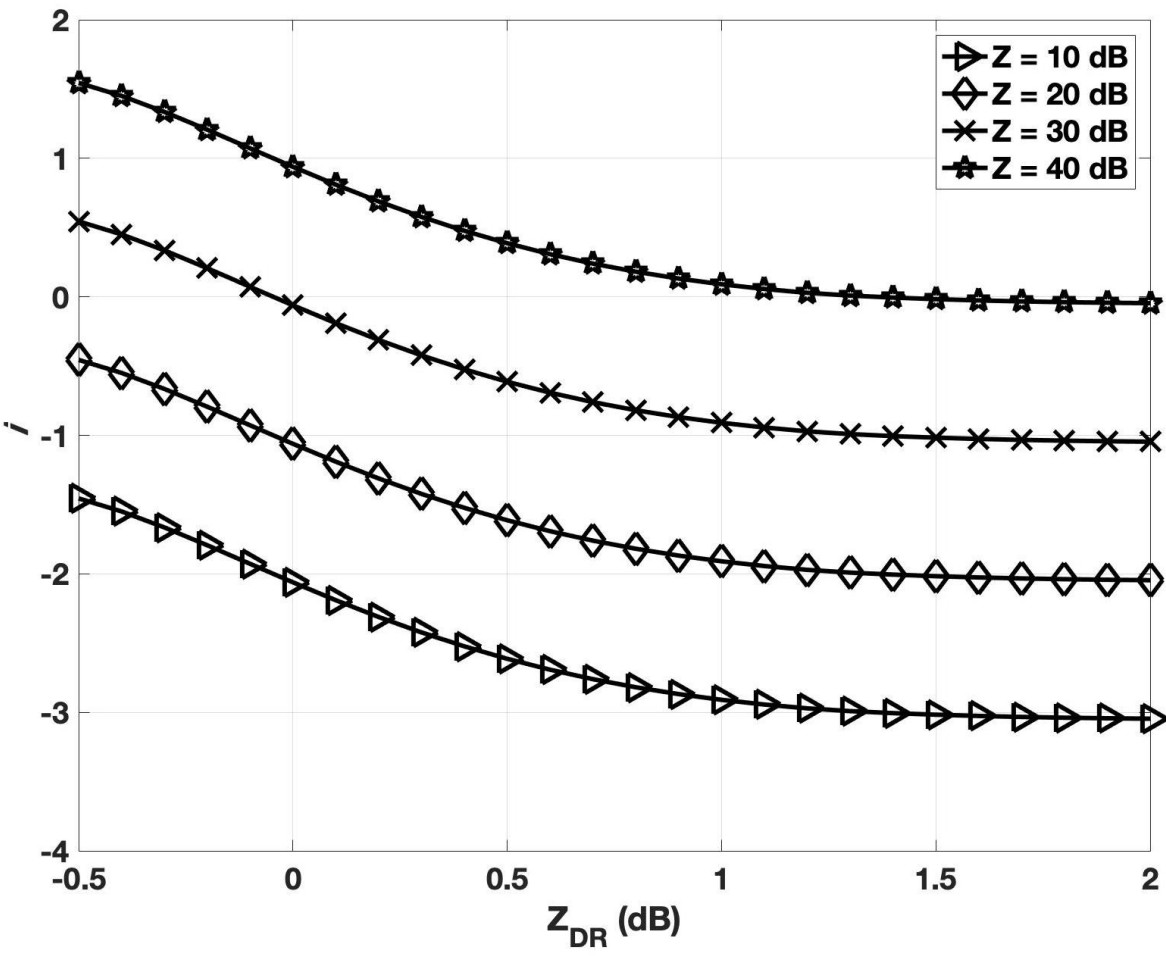

**Figure 11.** The calculated separation index respecting to different differential reflectivity values.

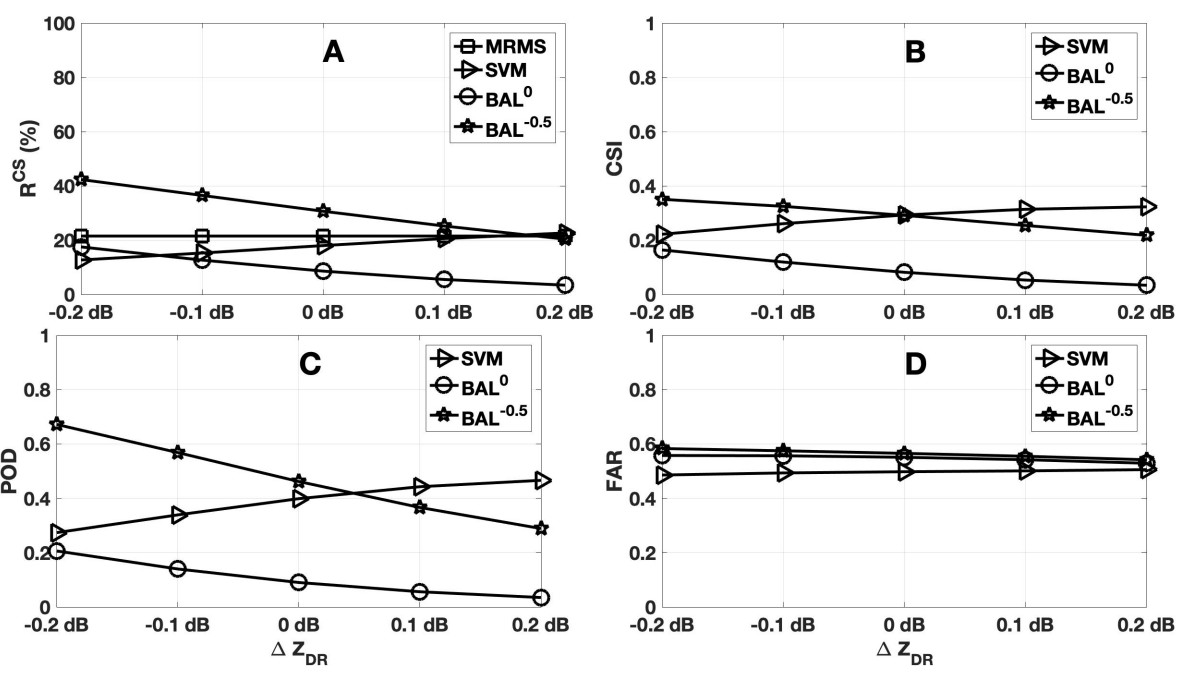

**Figure 12.** 24-hour averaged $R^{CS}$(A), CSI(B), POD(C), and FAR(D) from 30 August 2011. The results from BAL with threshold $T_0$ = -0.5, BAL with threshold $T_0$ = 0, SVM, and MRMS are indicated with symbols of pentagram, circle, triangle, and square, respectively.