# Peer review of "Separation of Convective and Stratiform Precipitation Using Polarimetric Radar Data with A Support Vector Machine Method"

_Atmospheric Measurement Techniques, 2019_

## Referee Comment (RC1) · Anonymous Referee #2 · 2 Jan 2020

Overall, it appears that the research was thorough and will add to the knowledge base of the field. Hence, I believe that after minor revisions, the paper should be published. However, there are some changes that I recommend to make the paper more readable and easier to understand:

*********************************************************************************** Line 2: "that require a whole volume radar data" should read as…. "that require a whole volume of radar data" Line 7: "with multiple precipitation events including two widespread mixture of stratiform and convective events" May read better as….. ""with multiple precipitation events including two widespread mixed stratiform and convective events"

[Figure]

line 10: "can accurately identify the convective cells from stratiform storms with the radar data only from the lowest scan in tilt. It can produce better results than using the separation index only." Would read better as....... "can accurately identify the convective cells from stratiform rain using radar data from the lowest scan tilt, and produced better results than using the separation index only." Line 15: "convective precipitation's are associated with" reads better as............ "convective precipitation is associated with" line 16: "while stratiform precipitations are associated" reads better as.......... "while stratiform precipitation is associated"

line 17: what is meant by saying a convective system consists of large and dense raindrops?

Line 53: "of the DSD-based approaches depends on the environmental regime"....Could you expand upon this a little bit please?

Line 58 and 59: "classification results even it is operated"...... Reads better as.... "classification results even if it is operated" line 68: "together with other three single polarization"....Reads better as... "together with the other three single polarization" line 73: "for convective and stratiform precipitations, total 4306 minutes of DSD data"....Reads better as...... "for convective and stratiform precipitation, a total of 4306 minutes of DSD data"

line 81: "stratiform precipitations generally consist of condense of small to median raindrops"..... This sentence needs to be corrected and explained better.

Line 98: "using the separation index i to identify convective precipitation from stratiform" ... This may read better as "using the separation index i to identify convective from stratiform precipitation"

Line 100: I assume that Nw refers to liquid water concentration... Is this true?

Equation 5: it may help to list as equations 5a) and 5b)

Lines 157 – 160: I understand the authors using the MRMS precipitation classification

algorithm as ground truth. . ..However, it should be noted that there are many imperfections in the system, especially since it only uses single pole information to determine echo classes

Line 165: "total one hour data were used as the convective type training data in the training data are associated with the >20 dBZs". . .. What is meant by "total one hour data"?

Line 169: " the number of support vectors is selected as 1000 and the current work". . ..Not everyone who reads this article will be familiar with some of the machine learning/artificial intelligence setting of criteria. . . It may help to add a few more lines on this. . . You have done that in lines 170 through 172 but still it would help to go a little bit further into what is typically done for these types of learning algorithms.

Line 188: "results from RCMK. . . . . .and MRMS could be different as large as five minutes in time stamps". . . . . . This may read better if written as "results from RCMK. . . . . .and MRMS could be significantly different with timestamp differences as large as five minutes" line 190: "evaluation criteria of the possibility of detection" . . . Should read as "evaluation criteria of the probability of detection" line 197: "was first validated with two widespread mixture of stratiform and convective". . . . . . Should read as. . . "was first validated with two widespread stratiform and convective mixed"

Line 241: "it could found that the heavy precipitation band". . . Should read as "it was found that the heavy precipitation band"

Line 253: "different from some existing classification algorithms" would read better as "different from other classification algorithms"

Lines 273 – 274: "second, the performance of the proposed approach highly depends on the training data. It should be very careful to select the training data." This would read better as "second, the performance of the proposed approach depends highly on the training data which should be very carefully selected. "

---

## Author Comment (AC1) · 9 Jan 2020

We do appreciate the reviewer provide so much important comments help us improving our manuscript. We'd like to address these comments as following.

Line 2: "that require a whole volume radar data" should read as….."that require a whole volume of radar data"

Response: the manuscript was modified following the reviewer's comment.

Line 7: "with multiple precipitation events including two widespread mixture of stratiform and convective events" May read better as…...with multiple precipitation events

including two widespread mixed stratiform and convective events"

Response: the manuscript was modified following the reviewer's comment.

line 10: "can accurately identify the convective cells from stratiform storms with the radar data only from the lowest scan in tilt. It can produce better results than using the separation index only." Would read better as........"can accurately identify the convective cells from stratiform rain using radar data from the lowest scan in tilt only, and produced better results than using the separation index only."

Response: the manuscript was modified following the reviewer's comment.

Line 15: "convective precipitation's are associated with" reads better as...convective precipitation is associated with"

Response: the manuscript was modified following the reviewer's comment.

line 16: "while stratiform precipitations are associated" reads better as.......while stratiform precipitation is associated"

Response: the manuscript was modified following the reviewer's comment.

line 17: what is meant by saying a convective system consists of large and dense raindrops?

Response: It was found the values of raindrop's mass weighted mean diameter(Dm) in stratiform and convective precipitation generally are within 1-1.9 mm and above 1.9 mm, respectively (Chang et al., 2009). Following the review's comment, we added more discussions in the manuscript L:73~76

Line 53: "of the DSD-based approaches depends on the environmental regime"....Could you expand upon this a little bit please?

Response: The separation index derived from Equations 2~5 in the manuscript depends on several factors: the radar wavelength, temperature, drops size distribution(DSD), and drop shape relations(DSR). The last three factors depend on environmental regime. In our work, we demonstrated that temperature, DSD and DSR features from Taiwan very similar to Darwin, Australia. Therefore, all the coefficients derived by BAL can be directly used in the current work. Following the review's comment, we added more discussions in the manuscript L:105∼112.

Line 58 and 59: "classification results even it is operated"......Reads better as....classification results even if it is operated"

Response: the manuscript was modified following the reviewer's comment.

line 68: "together with other three single polarization"....Reads better as...together with the other three single polarization"

Response: the manuscript was modified following the reviewer's comment.

line 73: "for convective and stratiform precipitations, total 4306 minutes of DSD data"....Reads better as......for convective and stratiform precipitation, a total of 4306 minutes of DSD data"

Response: the manuscript was modified following the reviewer's comment.

line 81: "stratiform precipitations generally consist of condense of small to median raindrops".....This sentence needs to be corrected and explained better.

Response: this sentence is modified as in L:73∼76.

Line 98: "using the separation index i to identify convective precipitation from stratiform" ... This may read better as...using the separation index i to identify convective from stratiform precipitation"

Response: the manuscript was modified following the reviewer's comment.

Line 100: I assume that Nw refers to liquid water concentration....Is this true?

Response: Nw is the normalized number concentration in the gamma drop size distribution.

Equation 5: it may help to list as equations 5a) and 5b)

Response: the manuscript was modified following the reviewer's comment.

Lines 157 – 160: I understand the authors using the MRMS precipitation classification algorithm as ground truth....However, he should be noted that there are many imperfections in the system, especially since it only uses single pole information to determine echo classes

Response: We appreciate the reviewer point this out. In the revision, we made the following statements in the revision L:151∼160: "The performance of MRMS has been thoroughly evaluated for years for the quantitative precipitation estimation, flash flood monitoring, severe weather and aviation weather surveillance (e.g., Gourley et al., 2016; Smith et al., 2016), and also used as the benchmark and/or ground truth in many studies (e.g., Grecu et al., 2016; Skofronick-Jackson and Coauthors, 2017). It should be noted that, although the performance of MRMS is well accepted in weather research community, there may be some imperfections in this system, especially it only uses single-polarization variables to determine the precipitation type. Other observations, such as the accumulated rainfall amount measured by gauges may be another reference. However, biases in the gauge measurements and improper R(Z) relations may causes other uncertainties. Therefore, at the current stage, MRMS precipitation classification result is the best benchmark in the training and validation of the proposed algorithm. Moreover, since the MRMS classification results are derived from 4 S-band radars, it can be viewed as an independent reference."

Line 165: "total one hour data were used as the convective type training data in the training data are associated with the >20 dBZs".....What is meant by "total one hour data"?

Response: The radar data collected from 1030 UTC to 1130 UTC are used in the

training. Given the radar VCP, there are total 13 volume scans data are available. We modified sentence as shown in L:163.

Line 169: " the number of support vectors is selected as 1000 and the current work"....Not everyone who reads this article will be familiar with some of the machine learning/artificial intelligence setting of criteria...It may help to add a few more lines on this...You have done that in lines 170 through 172 but still it would help to go a little bit further into what is typically done for these types of learning algorithms.

Response: We do appreciate the reviewer point this out. We added more discussion in the revised manuscript in L:165~175.

Line 188: "results from RCMK......and MRMS could be different as large as five minutes in time stamps"......This may read better if written as....results from RCMK......and MRMS could be significantly different with timestamp differences as large as five minutes"

Response: the manuscript was modified following the reviewer's comment.

line 190: "evaluation criteria of the possibility of detection".....Should read as the evaluation criteria of the probability of detection"

Response: the manuscript was modified following the reviewer's comment.

line 197: "was first validated with two widespread mixture of stratiform and convective"...... Should read as....was first validated with two widespread stratiform and convective mixed"

Response: the manuscript was modified following the reviewer's comment.

Line 241: "it could found that the heavy precipitation band"....Should read as....it was found that the heavy precipitation band"

Response: the manuscript was modified following the reviewer's comment.

Line 253: "different from some existing classification algorithms" would read better as "different from other classification algorithms"

Response: the manuscript was modified following the reviewer's comment.

Lines 273 – 274: "second, the performance of the proposed approach highly depends on the training data. It should be very careful to select the training data." This would read better as "second, the performance of the proposed approach depends highly on the training data which should be very carefully selected. "

Response: the manuscript was modified following the reviewer's comment.

––––––––––––––––––––––––––––

---

## Referee Comment (RC2) · Anonymous Referee #3 · 26 Apr 2020

This work proposes a new method, based on Support Vector Machine (SVM), to discriminate between convective and stratiform precipitation events. The algorithm receives radar data as input, namely the horizontal reflectivity, the differential reflectivity and the separation index. The results, presented in Section 3, highlight that the performance of the novel method are comparable with the multi-radar-multi-sensor (MRMS) precipitation classification approach, which was used as ground-truth. As a general comment, the manuscript is well structured and is adequate for the audience of AMT journal. However, before considering this work for publication, the authors must address some issues that are listed below:

- First of all, an important comment the proposed methodology, which uses the lowest unblocked scanning tilt, as stated by the authors at page 2 (Line 55). In my opinion, the authors should add a discussion about the weaknesses of such approach, considering, for example, the scenario in which it is applied in a complex-orography area. In such a case, the strategy may be not suitable, because the radar signal at lowest tilt may be totally or partially obstructed by the surrounding topography in some sectors. A possible solution to overcome this issue may be using the lower "free" available scanning elevation but this choice can generate inconsistencies and biases. For example, is some sectors of radar coverage, the algorithm may receive as input the reflectivity data collected at 1° elevation, in others the measurements sampled at 4° antenna elevation angle. The information provided by data sampled at 1° and 4° antenna elevation angles can be very different, depending on the precipitation type event that is taking place. - In Section 2, I suggest to add a figure showing the scanning geometry of the C-band polarimetric radars involved in this study. Please indicate the elevations angles used to develop the SVM method. Moreover, it is not clear if the authors used also the measurements provided by S-band single-polarization systems operating in the area of Taiwan. - In Section 2, the authors describe the variables used as input to the SVM method. They discuss about quality control of reflectivity measurements, focusing only on a specific issue, the attenuation along the path. I suggest to extend this discussion to other radar impairments that may have a strong impact on the performance of the proposed methods, such as the ground clutter (which strongly affects the radar measurements quality at lowest tilt) and the reflectivity vertical profile. In this respect, a detailed discussion should be provided about the bright band, which is a typical signature of stratiform precipitation events. - Section 2.3: in my opinion, it may useful cite some previous works that developed machine-learning algorithm based on meteorological radar data. I suggest the following references: Capozzi et al. (2018), Aditya Sai Srinivas et al. (2019) and Yen et al., (2019). - As training data for convective precipitation type, the authors use the measurements collected in a single event occurred on 23 July 2014. More specifically, for this event radar data collected from 10:30 to 11:30

(one hour) were used. I am quite skeptical about this choice, that the authors must justify and explain. It is well note that convective events may be triggered by different meteorological scenarios and that may exhibit different features in radar data according to thunderstorm types (single cell, squall line, supercell, etc.). Moreover, at page 6 (line 166) the authors declare that 17281 sets of data have been used in the training process. What does it mean "sets"? A clarification about this point is required. - In section 3, the authors present the results of their work, introducing a whole coverage convective ratio (RCS) number. The latter is defined as a parameter that provides a qualitative assessment of the performance of SVM and other considered methods. In my opinion, an evaluation about the reliability of SVM algorithm based on a single parameter is not sufficient to reach robust conclusions. Therefore, I suggest to involve in the statistical analysis other useful scores, such as the Critical Success Index and ROC curve. - Some suggestions about figures. In figure 1, I suggest to include a reference scale for terrain elevation. In figure 3, it is necessary to improve the line-style used to indicate the various algorithms. More specifically, MRMS and SVM time series seem have a similar marker according to the legend showed in panel (a). Regarding figure 4, I recommend to enlarge the panels, if it is possible. Moreover, the color scale should not have a gradient, because the output of the algorithm is binary (convective or stratiform). About Figures 5, 6 and 7, please clarify in the caption the meaning of black, red and white circles. Finally, I suggest to carefully checking the paper to address some minor typos.

**Best regards.**

**List of suggested references**

- Capozzi, V.; Montopoli, M.; Mazzarella, V.; Marra, A.C.; Roberto, N.; Panegrossi, G.; Dietrich, S.; Budillon, G. Multi-Variable Classification Approach for the Detection of Lightning Activity Using a Low-Cost and Portable X Band Radar. Remote Sens. 2018, 10, 1797.

**C3**

- Yen, M., Liu, D., Hsin, Y. et al. Application of the deep learning for the prediction of rainfall in Southern Taiwan. Sci Rep 9, 12774, 2019, https://doi.org/10.1038/s41598-019-49242-6.

- T., ASS, Somula, R, K., G, Saxena, A, A., PR. Estimating rainfall using machine learning strategies based on weather radar data. Int J Commun Syst. 2019;e3999. https://doi.org/10.1002/dac.3999.

---

## Referee Comment (RC3) · Anonymous Referee #4 · 11 May 2020

This is an interesting paper, that is well written. The authors specifically call this a prototype approach and name some limitations in the conclusion. The results of this machine learning approach are convincing, but I miss some discussion which would help to present a clearer picture to me. If those aspects are addressed, this work can be published.

ZDR is a moment that needs to be calibrated. How stable is the ZDR calibration with time for the C-Band you are using. Usually one attempts to be within +/- 0.2 dB. Do you use birdbath scans to calibrate ZDR? How sensitive is the separation index (eq2) to a ZDR bias? Are radome effects an issue (especially for the typhoon case you present; is

it possible that part of the somewhat unusual ZDR pattern in Fig 10 may be attributed to such a source?)?. You assume implicitly a perfect radar (hardware wise), where only attenuation corrections need to be applied (if necessary). I wonder how sensitive your method is to some radar hardware influences or issues. Or can you rule out any influence from the radar hardware? A discussion is needed here.

l 164 can you motivate why using such a large rhohv (> 0.98) as a criterion? You seem to throw away a lot of data e.g. if you have mixed phase precipitation with hail. Is there no hail in Taiwan? How much of the data are not considered? What happens if you observe rhohv < 0.98? How is the performance degrading if you have data ranges present that where considered for training. Those rangebins cannot be classified, since you trained the data for only specific ranges? Explain what consequence this choice of threshold has, how sensitive your results are, and before that, how the training results are dependent on this choice. Did you make sensitivity studies? L 166: what is exactly a "data set"? A range bin with all the moments you use satisfying the criteria for Z, RHOHV? Would be helpful to the reader who is not so familiar with this method. L 234: the intrinsic ZDR for stratiform precipitation: isn't it something around 0.2 dB. . .. Or is this different in Taiwan? Fig 10: ZDR looks biased to me. . .. There seem sector based (az range) biases for 270 - > 90°. . . . You mention this in l 250 ff, but Z looks relatively reasonable here.

---

## Author Comment (AC3) · 7 Jun 2020

**Reply to Referee 4**

We appreciate the reviewer provided these important comments help us improving our manuscript. We'd like to address these comments as following.

1. ZDR is a moment that needs to be calibrated. How stable is the ZDR calibration with time for the C-band you are using. Usually one attempts to be within +/- 0.2 dB. Do you use birdbath scans to calibrate ZDR?

**Response**: Thank you for the reviewer pointing this out. We totally agree with the reviewer that calibration plays a critical role in radar data processing and weather radar applications. A bias within 0.2 dB is the basic requirement on the ZDR field. In the current work, we directly used the data provided by the radar engineers from Central Weather Bureau of Taiwan, and no further calibration was applied on the $Z_{DR}$ field. We believed the quality of data is good, and the calibration bias of ZDR should be within the reasonable range based on following two reasons:

1.) This radar belongs to Weather Wing of the Chinese Air Force (CAF), and the data became available to the Central Weather Bureau (CWB) since 2009. Currently, RCMK is one of the operational radars in the radar network, and its data are used in the real-time quantitative precipitation estimation (QPE) and forecasting (QPF). The quality of the radar data is closely examined by the engineers from CAF and CWB. Therefore, we believe this radar is well maintained and calibrated.

2.) Same data sets (such as: 08/06/2009 ~ 08/09/2009) from this radar were also examined in few QPE papers (e.g., Wang et al. 2013, 2014). In order to achieve less than 10% bias in QPE products, the bias (including mis-calibration and attenuation) of reflectivity, and differential reflectivity should be within 1 dBZ, and 0.1 dB, respectively. Based on the QPE results estimated from this radar using different combinations of polarimetric radar variables, we believe the bias of Z and ZDR should be within a reasonable range.

On the other hand, following the reviewer's suggestion, we did the sensitivity analysis on the ZDR field. In this analysis, the observed ZDR field was manually adjusted by a factor of -0.2 dB, -0.1 dB, 0.1 dB, and 0.2 dB, respectively. The separation index was recalculated with the "biased" $Z_{DR}$ field. The performances from proposed approach and using separation index only were analyzed with the "biased" fields. Please refer to the reply to comment 2 for more details related to this test.

2. How sensitive is the separation index (eq2) to a ZDR bias? You assume implicitly a perfect radar (hardware wise), where only attenuation corrections need to be applied (if necessary). I wonder how sensitive your method is to some radar hardware influences or issues. Or can you rule out any influence from radar hardware? A discussion is needed here.

**Response**: First, we do appreciate the reviewer pointing this out. We did not include sensitivity analysis in the original manuscript. We believe such analysis is very useful to guide readers to evaluate and apply this algorithm.

To address this concern, we did the sensitivity test through simulation and real data validation. In the simulation part, the separation index $i$ was calculated with four distinct $Z$ values: 10 dBZ, 20 dBZ, 30 dBZ, and 40 dBZ. For each $Z$, $ZDR$ changes between -0.5 dB to 2 dB, which is used to simulate the bias on ZDR field. The simulation results could be found from revised manuscript in section 3.3.

In the real case validation, we did the following test:

1.) After correcting the $Z_{DR}$ field from attenuation, we manually added $\Delta Z_{DR}$ values (as the designed bias) on the corrected $Z_{DR}$ field. The $\Delta Z_{DR}$ values are: -0.2 dB, -0.1 dB, 0 dB, 0.1 dB, and 0.2 dB, and the "biased" $Z_{DR}$: are calculated as:

$$Z_{DR}^b = Z_{DR} + \Delta Z_{DR}$$

where $Z_{DR}^b$ indicates biased $Z_{DR}$.

2.) Calculate the separation index ($i^b$) with $Z_{DR}^b$. Evaluate the impacts of $\Delta Z_{DR}$ on performances of BAL$^0$ and BAL$^{-0.5}$ on cases 08/30/2011 and typhoon case (08/06/2009~ 08/09/2009).

3.) With $Z_{DR}^b$ and $i^b$ as the inputs to the proposed SVM approach, Evaluate the impacts of $\Delta Z_{DR}$ on performances of SVM approach on cases 08/30/2011 and typhoon case (08/06/2009~ 08/09/2009).

More details about simulation and real data validation could be found in section 3.2 in the revised manuscript. In the revised manuscript, only the case from 08/30/2011 is provided. The results from 2009 are provided as below:

[Figure]

**Figure 1.** 96-hour averaged $R^{CS}$(A), CSI(B), POD(C), and FAR(D) from 6~9 August 2009. The results from BAL with threshold $T_0$ = -0.5, BAL with threshold $T_0$ = 0, SVM, and MRMS are indicated with symbols of pentagram, circle, triangle, and square, respectively.

Are the radome effects an issue (especially for the typhoon case you present; is it possible that part of the somewhat unusual ZDR pattern in Fig. 10 may be attributed to such a source?)

**Response**:  Yes, we agree with the reviewer. The wet radome could be a possible issue for radar variables such as $Z$ and $Z_{DR}$. In the revised manuscript, we added following discussion:

Line 280 Other reasons such as wet radome may also contribute to the Z and $Z_{DR}$ issues.

L164, can you motivate why using such a large rhohv (> 0.98) as a criterion? You seem to throw away a lot of data e.g., if you have mixed phase precipitation with hail. Is there no hail in Taiwan? How much of the data are not considered? What happens if you observe rhohv < 0.98. How is the performance degrading if you have data ranges present that where considered for training. Those rangebins cannot be classified, since you trained the data for only specific ranges? Explain what consequence this choice of threshold has, how sensitive your results are, and before that, how the training results are dependent on this choice. Did you make sensitivity studies?

**Response**:  We'd like to address the reviewer's concern from following few different aspects:
1.)  In the manuscript, we use 0.98 as the threshold of RhoHV only in the training data selection. As reported by Kumjian (2013), pure rain generally produces very high of RhoHV (> 0.98) observed by WSR-88D. Such value (0.98) also suggested by Ryzhkov and Zrnic (2004) as the RhoHV field from majority of pure rain in C-band. Such large RhoHV was also suggested in hydrometeor classifications (e.g., Liu and Chandrasekar 2000; Park et al. 2009). For example, Park et al. (2009) suggested that RhoHVs for light/moderate rain, and heavy rain are 0.97 and 0.95, respectively. The precipitation may be classified as the mixed rain and hail if RhoHV is below 0.9. Following these pioneering works, we choose 0.98 as the threshold of RhoHV in the training data selection.

In the revised manuscript, we added the reference paper on Line 186.

2.)  The threshold of 0.98 for RhoHV is only applied in the training data selection. Such aggressive threshold can assure the training data from pure precipitation, and not smeared by clutter (including ground clutter, sea clutter, biological scatter), AP, and possible ice phase precipitation. When we test the algorithm with precipitation events, the threshold for RhoHV is selected as 0.90. Any pixel (gate) with RhoHV below than 0.9 is classified as non-precipitation echo. Any pixel with RhoHV above 0.9 is treated as pure rain, and the same support vector obtained from training data is applied.

3.)  The separation index ($i$) was derived from two drop size distribution (DSD) parameters $N_w$ and $D_0$. Therefore, it only validates at liquid phase precipitation (stratiform and convective types) as suggested by (Bringi et al. 2009). For other phase precipitation, such as mixed hail and rain, its performance is not well studied (Bringi et al. 2009). Other hydrometeor classification schemes are suggested for such scenario (Bringi et al. 2009). In this work, the separation index

also plays an important role in the SVM approach, therefore, we limited the application of the proposed approach only within pure water phase precipitation. We have not tested it on the mixed phase precipitation with hail. In the revised manuscript, we emphasized this limitation at Line 344.

4.) The goal of this work is to propose a prototype algorithm, and this manuscript focuses on describing this algorithm. We are working on further analyzing this approach including deriving the new separation index for S-band radar (WSR-88D), validating its long-term performance, including more variables (such as reflectivity texture), including multiple elevation angles. Sensitivity test for different training data definitely is also included in this work. We plan to report further findings in the upcoming papers.

L166, what is exactly a "data set"? A range bin with all the moments you use satisfying the criteria for Z, RhoHV? Would be helpful to the reader who is not so familiar with this method.

**Response**: We appreciate the reviewer pointing this out. A "set" means a set of data from one radar gate (defined as azimuthal angle and range). Be more specific, a set of training data means a vector of $[Z(a,r)\ Z_{DR}(a,r)\ i(a,r); d(a,r)]$. Where "$a$" indicates azimuthal angle, "$r$" indicates range; "$d$" is the desired response with "1" represents convective, and "-1" represents stratiform.

**Line 188:** A total of 17281 sets of data (15144 sets of stratiform, and 2137 sets of convective) are used in the training process. In this work, one data set is defined as the variables from a single gate in terms of range and azimuthal angle. Be more specific, a collection of training data means a vector of $[Z(a, r)\ Z_{DR}(a, r)\ i(a, r)\ d(a, r)]$, where $a$ and $r$ indicate azimuthal angle and range, respectively. The variable $d$ is the ground truth (with 1 and -1 represents convective and stratiform), i.e., the desired response in the training process.

L234: the intrinsic ZDR for stratiform precipitation: isn't it something around 0.2 dB, Or is this different in Taiwan?

**Response**: Yes, the reviewer is correct. The ZDR values we provided in the manuscript is not accurate. The ZDR values mentioned in the manuscript are within the black circle in the following figure (Fig. 7 in the original manuscript). If we examine carefully, especially for those gates with Z around 30 dBZ, the ZDR values are around 0.2 dB, instead of 0 dB.

[Figure]

Fig 10: ZDR looks biased to me… There seem sector based (az, range) biases for 270 -> 90⁰…
you mention this in l250 ff, but Z looks relatively reasonable here.

**Response**: We agree with the reviewer. In this sector, ZDR looks over corrected from attenuation, but Z looks relatively better. One hypothesis is both coefficients $\alpha$ and $\beta$ used in the linear PhiDP are need be adjusted based on the DSD and DSR features. Comparing to $\alpha$, $\beta$ is more sensitive to the impact of DSD and DSR.

**Reference:**

Kumjian, M. R., 2013: Principles and applications of dual-polarization weather radar. Part I: Description of the polarimetric radar variables. *J. Operational Meteor.,* **1** (19), 226-242, doi: http://dx.doi.org/10.15191/nwajom.2013.0119.

Liu, H., and V. Chandrasekar, 2000: Classification of hydrometeors based on polarimetric radar measurements: Development of fuzzy logic and neuro-fuzzy systems, and in situ verification. *J. Atmos. Oceanic. Technol.*, **17**, 140-164.

Park, H, A. V. Ryzhkov, D. S. Zrnic, and K.-E. Kim, 2009: The hydrometeor classification algorithm for the polarimetric WSR-88D: description and application to an MCS. *Weather and Forecasting*, **24**, 730-748.

Ryzhkov A. and D. Zrnic, 2004: Radar polarimetry at S, C, and X bands: comparative analysis and operational implications. *32$^{nd}$ Conference on Radar Meteorology*. 9R.3 24~29 May 2004

Wang, Y., P. Zhang, A. V. Ryzhkov, J. Zhang, and P.-L. Zhang 2014: Utilization of specific attenuation for tropical rainfall estimation in complex terrain. *Journal of Hydrometeorology*, vol 15, 2250-2266.

Wang, Y., J. Zhang, A. Ryzhkov, and L. Tang, 2013: C-band polarimetric radar QPEs based on specific differential propagation phase for extreme typhoon rainfall. *J. Atmos. Oceanic Technol.*, vol 30, 1354-1370..

---

## Author Comment (AC2)

**Reply to Referee 3**

We do appreciate the reviewer provide so much important comments  help us improving our manuscript. We'd like to address these comments as following.

1.) First of all, an important comment the proposed methodology, which uses the lowest unblocked scanning tilt, as stated by the authors at page 2 (Line 55). In my opinion, the authors should add a discussion about the weakness of such approach, considering, for example, the scenario in which it is applied in a complex-orography area. In such a case, the strategy may be not suitable, because the radar signal at lowest tilt may be totally or partially obstructed by the surrounding topography in some sectors. A possible solution to overcome this issue may be using the lower "free" available scanning elevation but this choice can generate inconsistencies and biases. For example, in some sectors of radar coverage, the algorithm may receive as input the reflectivity data collected at $1°$ elevation, in others the measurements sampled at $4°$ antenna elevation angle. The information provided by data sampled at $1°$ and $4°$ antenna elevation angle can be very different, depending on the precipitation type event that is taking place.

**Response**:  Thank you for the reviewer pointing this out. First, we totally agree with the reviewer that a discussion about the weakness of the proposed approach is necessary, which can guide readers to evaluate and implement this approach. We added following discussion in the revised manuscript:

1.) **Line 56**: Different from some existing classification techniques that require whole volume scan of radar data, this new approach uses the lowest unblocked tilt data in the separation. If the lowest tilt is partially or completely blocked, then next adjacent unblocked tilt is used instead.

2.) **Line 336**: Limitations of proposed approach are also included in the discussion section as: First, this approach is developed for fast scanning and fast update purpose, therefore, data from the lowest unblocked tilt is used as the input. However, if the radar is located in a complex orography area, radar beam could be partially or completely blocked at some regions. A possible solution for such scenario is using a hybrid scan data from different scanning tilts as the input. Radar scanning tilts used in the hybrid scanning are determined by the radar scanning geometry. Given the factor that precipitation's microphysics (such as drop size distribution) from different altitudes may be significantly different, therefore, the performance of proposed approach may worse than expected.

Secondly, the data from $1.4°$ elevation angle is used in the current work. Following figures show the scanning geometry of RCMK, and this figure was also added in the manuscript as the reviewer suggested. From this figure, we can find that the data from $0.5°$ is severely blocked by the central mountain range. Therefore, data from $1.4°$ elevation angle (treated as lowest unblocked data) is used in the current work.

[Figure]

Figure 1. Blockage maps of RCMK from the first 2 elevation angles (0.5° and 1.4°). The grey scale indicates the blockage percentages.

2.) In Section 2, I suggest to add a figure showing the scanning geometry of the C-band polarimetric radars involved in this study. Please indicate the elevations angles used to develop the SVM method. Moreover, it is not clear if the authors used also the measurements provided by S-band single-polarization systems operating in the area of Taiwan.

**Response**: Following the reviewer's suggestion, a figure of the scanning geometry of RCMK is added in the revised manuscript. The data from 1.4° elevation angle (the lowest unblocked tilt) is used in the algorithm development. We included this clarification in the revised manuscript. The S-band single-polarization radar data is not used in the SVM approach. We clarify this too in the revised manuscript.

3.) In Section 2, the authors describe the variables used as input to the SVM method. They discuss about quality control of reflectivity measurements, focusing only on a specific issue, the attenuation along the path. I suggest to extend this discussion to other radar impairments that may have a strong impact on the performance of the proposed methods, such as the ground clutter (which strongly affects the radar measurements quality at lowest tilt) and the reflectivity vertical profile. In this respect, a detailed discussion should be provided about the bright band, which is a typical signature of stratiform precipitation events.

**Response**: Following the reviewer's suggestion, issues about ground clutter and VPR are discussed in the revised manuscript. The discussion about bright band is also included as suggested.

**Line 98**: Other quality control issues, including calibration, reflectivity vertical profile, and

ground clutter removal, were also considered in this work. Since this radar is used in the real-time quantitative precipitation estimation, the biases of $Z$ and $Z_{DR}$ should be within 1 dBZ and 0.1 dB, respectively. The data quality of RCMK was examined through validating the QPE performance in different works (e.g., Wang et al., 2013, 2014). Therefore, the calibration bias of RCMK should be within a reasonable range. A vertical profile of reflectivity (VPR) correction is generally needed on the reflectivity field to reduce the measurement biases because of the melting layer (Zhang et al., 2011). Given the fact that $1.4°$ elevation angle is used within the maximum range of 150 km, and the melting layer is usually around 5 km in Taiwan, the radar data is well below the melting layer. In addition, considering the vertical profile of differential reflectivity is not well studied in the current stage, no vertical corrections are applied to fields of $Z$ and $Z_{DR}$. Ground clutter is typically associated with a low correlation coefficient ($\rho_{HV}$), the $\rho_{HV}$ threshold used in this work is 0.9, which can effectively remove those non-meteorological echoes such as ground clutter.

**Line 163**: On the other hand, stratiform precipitations are generally associated with a prominent bright band signature. The melting hydrometeors increase backscatter during stratiform rainfall, which can significantly enhance radar reflectivity. The bright band feature is one of the obvious indicators of stratiform precipitation. Bright band signature normally can be observed from relatively high EAs (such as above $9.9°$). From low EAs, because of the combination of radar beam broadening and low slant angle, the bright band feature spreads into more gates and becomes not apparent. Therefore, in this work, the bright band feature from high elevation angles is only used in training data selection but not used as one of the inputs.

4.) Section 2.3, in my opinion, it may useful cite some previous work that developed machine-learning algorithm based on meteorological radar data. I suggest the following reference: Capozzi et al. (2018), Adity Sai Srinivas et al. (2019) and Yen et al. (2019).

**Response**: Following the reviewer's suggestion, these three references were added into the revised manuscript.

Line 139: Machine learning algorithms based on meteorological radar data were well developed during the past two decades (e.g., Capozzi et al., 2018; T. et al., 2019; Yen et al. 2019)

5.) As training data for convective precipitation type, the authors use the measurements collected in a single event occurred on 23 July 2014. More specifically, for this event radar data collected from 10:30 to 11:30 (one hour) were used. I am quite skeptical about this choice, that the authors must justify and explain. It is well note that convective events may be triggered by different meteorological scenarios and that may exhibit different features in radar data according to thunderstorm types (single cell, squall line, supercell, etc.).

**Response**: In this work, the training data plays a critical role in the SVM development. Therefore, we choose convective and strtiform precipitations following three major steps.

1.) First, the training data was checked following general classification principles: for example, heavy precipitation band associated with high reflectivity for convective type precipitation; bright band for stratiform type precipitation.

2.) Second, the ground observation is used as another reference. For example, the severe weather report could be used as the ground observation.

3.) The classification results from MRMS is used as the third reference.

The convective type precipitation data is mainly from a thunderstorm on 23 July 2014. An aircraft crash tragedy caused by strong downdraft is used as the ground observation. MRMS classification algorithm classifies this event as the convective precipitation type. The radar observation of reflectivity $Z$ and differential reflectivity $Z_{DR}$ at 0858 UTC is shown in Fig. 2

[Figure]

Figure 2. Reflectivity (left) and differential reflectivity (right) at 0858 UTC, 23 July 2014.

A clear squall line features can be identified at this moment, which triggered the strong updraft/downdraft. Inside this squall line, the reflectivity field is above 40 dB; differential reflectivity field is above 1 dB. The maximum value of $Z_{DR}$ could be as high as 2.5 dB. Behind the severe precipitation band, the differential reflectivity field drops to negative value because of the attenuation issue. Fields of $Z$ and $Z_{DR}$ from 1028 UTC are shown in Fig. 3. Although the squall line signatures are not as well structed as 0858 at this moment, clear convective precipitation features such as large reflectivity, and very positive differential reflectivity are still very obvious. Therefore, we use those gates classified as convective type as in the training data.

We hope these plots can address the reviewer's concerns.

[Figure]

Figure 3. Reflectivity (left) and differential reflectivity (right) at 1028 UTC, 23 July 2014.

6.) Moreover, at page 6 (line 166) the authors declare that 17281 sets of data have been used in the training process. What does it mean "sets"? A clarification about this point is required.

**Response**:  We appreciate the reviewer pointing this out. A "set" means a set of data from one radar gate (defined as azimuthal angle and range). Be more specific, a set of training data means a vector of $[Z(a,r)\ Z_{DR}(a,r)\ i(a,r); d(a,r)]$. Where "$a$" indicates azimuthal angle, "$r$" indicates range; "$d$" is the desired response with "1" represents convective, and "-1" represents stratiform.

**Line 188:** A total of 17281 sets of data (15144 sets of stratiform, and 2137 sets of convective) are used in the training process. In this work, one data set is defined as the variables from a single gate in terms of range and azimuthal angle. Be more specific, a collection of training data means a vector of $[Z(a, r)\ Z_{DR}(a, r)\ i(a, r)\ d(a, r)]$, where $a$ and $r$ indicate azimuthal angle and range, respectively.   The variable $d$ is the ground truth (with 1 and -1 represents convective and stratiform), i.e., the desired response in the training process.

7.) In section 3, the authors present the results of their work, introducing a whole coverage convective ratio (RCS) number. The latter is defined as parameter that provides a qualitative assessment of the performance of SVM and other considered methods. In my opinion, an evaluation about the reliability of SVM algorithm based on a single parameter is not sufficient to reach robust conclusions. Therefore, I suggest to involve in the statistical analysis other useful scores, such as the Critical Success Index and ROC curve.

**Response**:  We agree with the reviewer that a single crietia may not sufficient to validate the performance of proposed approach. To address the reviewer's concerns, we made the following modifications:

1.) Besides the convective ratio ($R^{CS}$) we introduced in the original manuscript, we also applied the Probability of Detection (POD), False Alarm Ratio (FAR), and Critical Success Index (CSI) in the performance evaluation.

2.) Since both cases of 30 August 2011 and 14 June 2012 are widespread stratiform and convective mixed precipitation events, and the performances of proposed approach show similarity from these two cases. We only kept the 30 August 2011 cases in the revised manuscript for the stratiform and convective mixed precipitation case. We also added more analysis and sensitivity test on this case.

3.) For the tropical precipitation case 08/06/2009~08/09/2009 case, we included POD, FAR, CSI analysis, and also included sensitivity test.

Please refer section 3 in the revised manuscript for more details.

8.) Some suggestions about figures. In figure 1, I suggest to include a reference scale for terrain elevation.

**Response**:  Following reviewer's suggestion, a reference scale for terrain elevation is added in the manuscript, as shown below.

[Figure]

Figure 4. The terrain of Taiwan, the location of a C-band polarimetric radar RCMK (marked with a black square), JWDs (marked with black stars), and four S-band single polarization radar

RCCG, RCKT, RCHL, and RCWF (marked with black circles). The continuous grey-scale terrain map shows the central mountain range of Taiwan.

9.) In figure 3, it is necessary to improve the line-style used to indicate the various algorithm. More specifically, MRMS and SVM time series seem have a similar marker according to the legend showed in panel (a).

**Response**: Following reviewer's suggestion, we use different colors to represents the results from different algorithms. More details could be found from the response to comment 7.).

10.) Regarding figure 4, I recommend to enlarge the panels, if it is possible. Moreover, the color scale should not have a gradient, because the output of the algorithm is binary (convective or stratiform).

**Response**: Following reviewer's suggestion, we made following modifications: 1.) enlarge each panels in figure 4; and 2.) change the color scale as binary.

11.) About Figures 5, 6, and 7, please clarity in the caption the meaning of black, red, and white circles.

**Response**: Following reviewer's suggestion, we added the meaning of these circles in the caption.

12.) Finally, I suggest to carefully checking the paper to address some minor typos.
**Response**: Following reviewer's suggestion, we run grammar and spelling check before submitting the revision.

---

## Referee Report (RR1)

Review of "Separation of Convective and Stratiform Precipitation Using Polarimetric Radar Data with A Support Vector Machine Method" by Yadong Wang, Lin Tang, Pao-Liang Chan, and Yu-Shuang Tang

Thanks for the revised version of the manuscript. I see part of my comments and suggestions considered. Thank you. Thanks for carrying out the sensitivity study. I have the feeling that it may be better to remove this portion as it stands right now. In particular the scores for different ZDR bias indicates the best scores for the BAL -0.5 assuming a bias of ZDR = -0.2 dB version. So is this an indication, that there is actually a ZDR bias in the underlying data? See e.g Fig 10. So overall, the paper presents a new method that has potential, and should be published. But to me the results are not clear enough because an evaluation of the methods is complicated by the obvious data quality issues.

Also, I'm a bit confused about your choice of the rhohv thresholds. The reasoning and consequences on the method need to elaborated see also below.

Some probably minor revisions are needed.

I have some more comments as I go along the paper.

L 16: relatively low R: please provide a number to get an idea on what is considered a low T´R

L 18: state why the QPE accuracy is better if one makes a distinction into those two regimes.

L 40. "An alternative scanning scheme …." Something seems wrong in the sentences (grammar)

L 58: The use of lowest unblocked sweep is used. Don't you have to live with the lowest sweep, even if it is partially blocked. You don't have another sweep available, see l. 38

L 107: Threshold of 0.9 is too high to discriminate clutter and meteorology. You will loose valid meteorological data! I think I addressed this already in my initial review. You have to show or discuss why this choice doesn't affect your results. I would expect that you will have problems with your convective data set.

L 165: if you use a rhovh threshold of 0.9, you will exlude already a good part of the bright band data!

L 186: really a rhohv > 0.98?

L 205: would be good to state again what those threshold represent… actually, I tried to find the meaning of $T_0$ in the previous sections and I couldn't find it. Please clarify! What does a threshold of -0.5 physically actually mean?

Fig5: why do you consider only the area in the white circle? What about the rest?

L 265 ff: You rightly state that the potential data quality problems. As such, I wonder to what extent a comparison of the methods is really meaningful. Here the SVSM seems to outperform the the BAL methods, correct? On the other hand, the CSI is rather poor for all methods, correct?

L 285: why don't you show the scores from the stratiform event?

L 295: ZDR calibration bias….: plus the over correction due the attenuation correction, or? But I don't understand the reasoning that positive ZDR bias should lead to more stratiform classification.

---

## Editor Decision (ED1)

Dear Authors, It is great the effort done in improvement this work through its revisions and in my opinion most of comments of Reviewers have been addressed. Anyway, I retain that this work requires further improvements. First, language and text have to be improved. In this respect, I have noticed that several sentences do not appear sufficiently connected among them, with sentences and concepts that do not appear derived from the previous arguments. Moreover, several typos remain (for example lines 137, 146, 160). Second, I agree with the comment of Reviewer 1 on the quality of the datasets: only two meteorological events have been used for the training datasets, and the 17281 datasets are largely unbalanced towards stratiform data. I understand that this work deal of a prototype algorithm, and in this respect I could be acceptable so few events, anyway this limitation has to be clearly and extensively reported in the text (including §2.3.2, abstract and conclusions), and the indicated unbalance justified in some extent. Finally, I retain answers to Reviewers acceptable, but not fully addressed in the text. In other words, answers give a detailed justification to the points indicated by Reviews but the revised text in most cases addresses the point in a sentence not sufficiently complete. My general comment is to take answers to Reviewers and insert them in the text as much as possible "as is". In particular, I retain necessary the justifications for the following points:

- The selection of the datasets, as previously indicated;
- The explanation for the 11 mm/hour value, citing the two formulas and the threshold of 40 dBZ with its reference;
- The sentences on the QPE accuracy have to be revised according to the answer to Reviewer (in my opinion much clear than the sentence in the text), and the paper of Kirsch et al. 2019 have to be included in references;
- The justifications on the $\rho_{HV}$ values have to be absolutely inserted in the text, in particular points 1 and 2 of the answer to Reviewer that I suggest to include as much as possible "as is". On the other hand, I agree that the following discussion in the answer on the small differences using different thresholds have to be reported in the text but in a much more compact form. Anyway, the discussion on the $\rho_{HV}$, in the revised text is only in conclusions and in a form too concise, while it is absolutely necessary address properly this point , as done in the answer to the Reviewer;
- The sentence "In this work, we only use bright band […] to remove bright band signature" of the answer, has to be addressed better in the text;
- When presenting the two principal study cases, it is necessary explicitly declare that small circle areas will be used for further analysis, reporting the justification of the answer to the Reviewer;
- Even if obvious, the values of the statistical parameters of the pure stratiform case have to be reported for symmetry with the other two cases, as declared in the answer but not done;
- Finally, also the justification respect the Zdr calibration bias have to be included in the text.

All the Best.

---

## Author Response (AR2)

**Reply to Referee 1**

We appreciate the reviewer provided these important comments help us improving our manuscript. We'd like to address these comments as following.

First, as I read into the details, I saw in lines 182 through 184 that only one case was used to train the convective portion of the technique…. This seems very small… if this is true, how many cases (not just radar tilts) were used to train the stratiform portion of the SVM technique… This is one of the chief reasons I rated the scientific quality as only fair…. Otherwise, I would've rated it higher.

**Response**:
We appreciate the reviewer pointing this out. As we mentioned in the manuscript, convective type training data is mainly from a strong convective precipitation event on 23 July 2014. This thunderstorm is clearly identified as a convective precipitation through ground observation and MRMS results, that is the major reason we chose this case. Besides this case, some convective radar gates from the precipitation event on 30 August 2011 are also used as the convective training data. These convective radar gates are identified by the MRMS results. We clarified these in the revised manuscript. Total 17281 sets of data (including convective and stratiform types) are used in the training. We believe this is a reasonable number of training data at the stage of prototype algorithm development. High more numbers of training data and support vectors should be used if this approach is used for the operation purpose.

Also, the overall readability and the English needs to be improved significantly… The reason is there are statement written that will convey to the reader an opposite meaning than what the authors clearly intended. An example, can be found in line 227… where the authors mention that POD, FAR, CSI can not capture the performance of the SVM technique… If that were true why bother with those statistics …., However, I think the authors really meant that these statistics only partially capture the performance and therefore a new performance evaluation statistics should also be considered. The same subject was also mentioned in the conclusion in line 332 through 335 and should be clarified there as well… where it significantly caused me to wonder why the stats were even used …, Again, I know the authors did not intend that meaning…, but the way it is written conveyed that type of meaning…

**Response**:
Thank you. We totally agree with the reviewer that there are some inconsistent statements in the manuscript. In the revised manuscript, we double checked the statements, grammar, and other English issues, and corrected these errors.

We agree with the reviewer that those traditional statistics only partially capture the performance of the proposed approach. A new performance evaluation score is therefore used as a complement. We corrected this in the revised manuscript.

While I think the authors significantly improved the amount of detail needed for publication and that there is significant scientific merit to the work. I just do not think it's ready for publication yet without significant major revisions to improve the readability of the paper. However, I definitely think this paper should be revised because I believe it does carry information that would be useful for the scientific community once the revisions are made. While I know this may be discouraging for authors, they need to realize that important that they keep moving forward on this paper because persistence will ultimately in excellence.

**Response**:
We do appreciate the reviewer's comments, which helps us making this paper more solid. In the revised manuscript, we double checked the statements, grammar, and other English issues, and corrected these errors.

**Reply to Referee 2**

We appreciate the reviewer provided these important comments help us improving our manuscript. We'd like to address these comments as following.

1. Thanks for carrying out the sensitivity study. I have the feeling that it may better to remove this portion as it stands right now. In particular the scores for different ZDR bias indicates the best scores for the BAL-0.5 assuming a bias of ZDR=-0.2 dB version. So is this an indication, that there is actually a ZDR bias in the underlying data?

**Response**: Following the reviewer's suggestion, the section of sensitivity analysis was removed in the latest version.

We agree with the reviewer that BAL$^{-0.5}$ shows the best CSI and POD scores when an artificial -0.2 dB bias was added on the ZDR field. However, we do not think we can affirm that the actual ZDR is biased (-0.5 dB) for the following two reasons:

1.) With the manually added negative ZDR biases, BAL$^0$ and BAL$^{-0.5}$ will classify more echoes as convective type. Based on the separation index calculation (Equations 2~ 5), it could be found that the separation index becomes larger with smaller ZDR value for a given reflectivity. As a result, radar echo associated with larger separation index is more likely be classified as convective type. Similar results also could be found in the simulation. That is the major reason that POD and CSI are significantly enhanced. However, on the other hand, we also can find that BAL$^{-0.5}$ with +0.2 dB ZDR bias produces better RCS score comparing to -0.2 dB ZDR bias. Therefore, +0.2 dB ZDR bias produce better RCS, and -0.2dB bias produce better CSI and POD. We can't say which one is better.

2.) On the other hand, ZDR is directly used as an input in the SVM approach. From the sensitivity test (Figure 12), we can find that ZDR with +0.2 dB bias produce better scores (RCS, POD, CSI, and FAR) than -0.2 dB bias.

Since the ZDR value is directly or indirectly used in both approaches, and we got two opposite trends, we can't draw a convincible conclusion of positive or negative bias. We appreciate the reviewer emphasis the concerns about the ZDR bias, and we know the ZDR bias play an important role in the radar echo classification. When we apply this approach into operation, we need to be very careful about the data quality control. We included more discussion in the conclusion part.

2. L 16: relatively low R: please provide a number to get an idea on what is considered low T'R .

**Response**: Two typical R-Z relations are well used the rainfall rate estimation:
$Z = 200R^{1.6}$(stratiform), and $Z = 300R^{1.4}$ (convective). Given the fact that the reflectivity from a stratiform precipitation can't above 40 dBZ. Therefore, the maximum precipitation rate for stratiform precipitation is 11 m/hour.
Following the reviewer's suggestion, we added this value into the manuscript.

3.  State why the QPE accuracy is better if one makes a distinction into those regimes.

**Response**: Different reflectivity-rain rate relations are generally used in QPE (Kirsch et al. 2019). Therefore, accurate classification help choosing the optimum R(Z) relation, which can produce accurate rainfall estimation results.

Following the reviewer's comment, we added this in the revised manuscript.

L40. "An alternative scanning scheme ...." Something seems wrong in the sentences (grammar).

**Response**: This sentence is modified as:

"New radar scanning schemes are designed to update data from low elevations in a high frequency and data from high elevations in a low frequency. Such alternative scanning scheme enables the WSR-88D radars to promptly capture the storm development, which can enhance the weather forecast capability and QPE accuracy."

L58. The use of lowest unblocked sweep is used. Don't you have to live with the lowest sweep, even if it is partially blocked. You don't have another sweep available. See I38

**Response**:
If the lowest tilt data is partially/totally blocked, the reflectivity and differential reflectivity values become biases/unavailable. Therefore, we have to use the next adjacent low tilt data. We believe the sentence in L38 misleading, and we modified the sentence in the revised manuscript.

L107: Threshold of 0.9 is too high to discriminate clutter and meteorology. You will loose valid meteorological data. I think I addressed this already in my initial review. You have to show or discuss why this choice doesn't affect your results. I would expect that you will have problems with your convective data set.

L 186: really a rhohv > 0.98?

**Response**: These two comments are both related to the $\rho_{HV}$ thresholds in the SVM approach training and testing. We do appreciate the reviewer bringing this back to us, since this is a very important issue in the data quality control. We tested a new set of thresholds ($\rho_{HV}$ = 0.9 in training and $\rho_{HV}$ = 0.85 in quality control) in the algorithm and clarify this issue in the manuscript.

Following the reviewer's suggestions, we did the following modifications in our SVM approach.
1.) In the training data selection, a threshold of 0.9 is applied to replace 0.98. This threshold included more radar echoes in the training data. All parameters and coefficients used in the SVM approach are changed according to the new training data. These new settings were used in the validation cases. Following the work reported in Park et al. (2009), the lowest $\rho_{HV}$ from typical liquid phase precipitation (including light to heavy rains) is 0.92. Since the goal of this

work is proposing a liquid phase precipitation classification method, we believe 0.9 is a reasonable threshold in the training data selection.

2.) In the validation, a $\rho_{HV}$ threshold of 0.85 is used to remove those echoes not associated with rainfall. Radar echoes associated with $\rho_{HV}$ larger than 0.85 are remained in the classification. We understood that other meteorological targets such as wet snow, crystals, graupel may be associated with low $\rho_{HV}$ (< 0.85). However, this work proposes an approach to classify precipitations into either stratiform or convective types. Classification of other meteorological targets (snow, graupel) is not in the scope of this work. As reported in Park et al. (2009), the $\rho_{HV}$ threshold of 0.9 is good for the majority of the liquid phase precipitation. The $\rho_{HV}$ from the mix of rain and hail is also above 0.85. In this work, any pixel associated with low $\rho_{HV}$ (< 0.85) is set as null.

In the revised manuscript, Figures 4 ~ 10 were regenerated using the new thresholds ($\rho_{HV} = 0.98$ for training, and $\rho_{HV} = 0.9$ for quality control). Overall, the new thresholds do not significantly affect the results. A comparison of SVM results using old threshold set ($\rho_{HV} = 0.98$ for training, and $\rho_{HV} = 0.9$ for quality control), and new threshold set ($\rho_{HV} = 0.9$ for training, and $\rho_{HV} = 0.85$ for testing) is shown in Figure 1. In this example, we could find that major parts from these two threshold sets show similar classification results, and only slight differences could be found within the red circles.

Based on our analysis, similar results may come from two reasons:
First one is the overall high $\rho_{HV}$ field. These cases used in this work are from pure liquid precipitation events. No apparent hails contamination (within convective) could be identified from these cases. The model sounding data indicates that the bright band is about 5.5 km above the surface, which can eliminate the bright band effect. Therefore, high $\rho_{HV}$ field are generally observed from these cases. Therefore, the adjustment of the $\rho_{HV}$ threshold does not have significant effects on the classification results. An example of the $\rho_{HV}$ field is shown Fig. 2, where the $\rho_{HV}$ values from most parts of the fields are above 0.85.

Second reason may exist in the data processing. In the proposed approach, the reflectivity and differential reflectivity fields are also further smoothed with a 3 by 3 windows after attenuation correction. This step can further decrease the variations from some individual pixels. Therefore, although some individual pixel associated with low $\rho_{HV}$ is eliminate by threshold, the smoothing average still can fill in. To demonstrate this point, the reflectivity (Fig. 3), differential reflectivity (Fig. 4), and separation index (Fig. 5) fields processed using old threshold ($\rho_{HV} = 0.9$) and new threshold ($\rho_{HV} = 0.85$) are shown in panels A and B, respectively. Based on these comparisons, we can understand why the classification results from previous and new thresholds are similar.

Overall, we do appreciate the reviewer emphasis this issue again to us, which makes us go back to investigate the impacts of the $\rho_{HV}$ thresholds. We added more discussion about the $\rho_{HV}$ threshold selection. We also realize that different thresholds may cause other potential issues. In this work, we focus on proposing a classification approach. The statistical analysis about the performance of this approach including the sensitivity analysis will be reported in coming manuscripts.

[Figure]

Figure 1. the SVM results using: A) previous thresholds (0.98 and 0.9) and B.) new thresholds (0.9 and 0.85).

[Figure]

Figure 2. the $\rho_{HV}$ field.

[Figure]

Figure 3. the smoothed reflectivity using: A) previous thresholds (0.98 and 0.9) and B.) new thresholds (0.9 and 0.85).

[Figure]

Figure 4. the smoothed differential reflectivity using: A) previous thresholds (0.98 and 0.9) and B.) new thresholds (0.9 and 0.85).

[Figure]

Figure 5. the separation index using: A) previous thresholds (0.98 and 0.9) and B.) new thresholds (0.9 and 0.85).

L 165: if you use a rhohv threshold of 0.9, you will exclude already a good part of the bright band data?

**Response**: Yes, we agree with the reviewer that $\rho_{HV}$ threshold of 0.9 could take care of major part of the bright band. Moreover, we investigated the model sounding data for these cases. It was found that the $0^o$ temperature height is around 5 to 5.5 km in these cases. Given the fact of the low elevation angle (1.4°) and maximum range (150 km), all data is well below bright band. In this work, we only use bright band as one of the criteria to identify stratiform type precipitation, and $\rho_{HV} = 0.9$ will not applied to remove bright band signature.

L 205: Would be good to state again what those thresholds represent…. Actually, I tried to find the meaning of T_0 in the previous sections and I couldn't find it. Please clarify what does a threshold of -0.5 physically actually mean?

**Response**:
Using separation index only to classify precipitation into either stratiform or convective was originally proposed by Bringi et al. (2009). In their work, a threshold of 0 was suggested, and the radar echo from a gate could be classified as stratiform ($i < 0$) or convective ($i > 0$). In this work, we first tested the performance of BAL using 0 as the threshold, and it was found that a lot of radar echoes associated with convective features were classified as stratiform. Therefore, we tested a very aggressive threshold of -0.5, which classify more echoes as convective.

Following the reviewer's suggestion, the following sentence is added in the revised manuscript:

"The threshold of $T_0$ = 0 was suggested by BAL to classify precipitation into stratiform or convective types. In the current work, a more aggressive threshold of $T_0$ = -0.5 is also used in the evaluation, which will classify more radar echoes as convective than threshold of $T_0$ = 0."

Fig5: Why do you consider only the area in the white circle? What about the rest?

**Response**:
In this example, we can find that $BAL^0$ /$BAL^{-0.5}$ classifies the least/most echoes as convective, and SVM shows the best results comparing to BAL approach. The classification difference could be found from different locations. Here we only use the region within the circle to analyze why these three approaches show different results. Similar reasons may be applied to other regions. That is the reason that we did not discuss the region out of the circle.

L 265: you rightly state that the potential data quality problems. As such, I wonder to what extent a comparison of the methods is really meaningful. Here the SVM seems to outperform the BAL methods, correct? On the other hand, the CSI is rather poor for all methods, correct.

**Response**:
Thank you for reviewer pointing this out, this is a really good comment.
As we presented in the manuscript, using separation index in the precipitation classification is a good option in the fast scan fast update radar operation scheme. However, single separation index may be affected by the uncertainties in the radar field and DSD assumption. Therefore, the classification approach is potentially improved through integrating more variables. The motivation of this work is proposing a complementary method to enhance the performance of BAL, and we did identify enhancements in different precipitation events. Although scores of RCS, CSI, POD and FAR used in the manuscript show improvements in these cases, we still think these are still at the qualitative evaluation stage (Line 308 in the manuscript). These scores are rather used to demonstrate the advantages of proposed approach than statistical analysis. In order to get statistical results, we believe more long-term precipitation events are needed. The purpose of this work is introducing this SVM approach, and statistical analysis will be addressed in the future work.

We agree with the reviewer that the CSI scores are low for all methods during the comparison with MRMS. Two factors could be the major reasons: First, MRMS results are derived using the mosaicked field from four S-band radar, and the classification results are produced very 10 minutes, and other three approaches derived from RCMK, and the time gaps between MRMS and these three approaches could be as large as 5 minutes. Second, a convective storms' size, intensity, and convective cell locations could change dramatically during a short period. Therefore, pixel-to-pixel based evaluation scores may lose tracking the performance of the proposed approach, which makes the CSI showing low value. However, overall, the scores from SVM shows higher values comparing to BAL.

L 285: why don't you show the scores from the stratiform event.

**Response**:

This is a pure stratiform precipitation event. No convective cells are identified by all these approaches. The POD = 100%, CSI = 100%, and FAR = 0. Therefore, no scores and plots were shown in the manuscript. We added these scores in the revised manuscript.

L 295: ZDR calibration bias ….: plus the over correction due to the attenuation correction. But I don't understand the reasoning that positive zdr bias should lead to more stratiform classification.

**Response**:

The separation index is calculated using Z, ZDR and an DSD assumption (Equations 2~ 5). The calculated separation index shows larger value with a smaller ZDR value for a given reflectivity. The classification result is based on the comparison between calculated separation index and a threshold. A larger separation index is more likely be classified as convective type, and smaller separation index is more likely be classified as stratiform type. Therefore, when a positive bias is added into the ZDR field, the ZDR value becomes bigger, and the calculated separation index will show smaller values. As the results, more radar echoes will be classified as stratiform type.

**Reference:**

Kirsch, B., M. Clemens, F. Ament, 2019: Stratiform and convective radar reflectivity-rain rate relationships and their potential to improve radar rainfall estimation. *Journal of Applied Meteorology and Climatology,* vol 58, 2259-2271

LIST OF CHANGES MADE IN THE MAUSCRIPT

1.) Remove the sensitivity test section.
2.) Change the thresholds of $\rho_{HV}$ used in the training data selection and quality control. Regenerate all results using the new thresholds.
3.) Add more discussions required by the reviewers in the revised manuscript.
4.) Modify/rewrite sentences suggested by the reviewers.

[revised manuscript text omitted]

---

## Author Response (AR3)

**Reply to Referee**

We appreciate the reviewer provided these important comments help us improving our manuscript. We'd like to address these comments as following.

*First, language and text have to be improved. In this respect, I have noticed that several sentences do not appear sufficiently connected among them, with sentences and concepts that do not appear derived from the previous arguments. Moreover, several typos remain (for example lines 137, 146, 160).*

**Response**: We appreciated the reviewer's comments. We double checked language and text parts in the manuscript. Got helps from writing center and professional writing software. Issues in grammar, sentence, and word should be corrected.

*Second, I agree with the comment of Reviewer 1 on the quality of the datasets: only two meteorological events have been used for the training datasets, and the 17281 datasets are largely unbalanced towards stratiform data. I understand that this work deal of a prototype algorithm, and in this respect I could be acceptable so few events, anyway this limitation has to be clearly and extensively reported in the text (including §2.3.2, abstract and conclusions), and the indicated unbalance justified in some extent.*

**Response**: Following the reviewer's comment, the limitations of the training data, including the total number and stratiform/convective ratio are clearly explained in abstract, section 2.3.2, and conclusions.
**In abstract**: The weight vector and bias in the support vector machine were optimized using well-classified data from two precipitation events.
**In section 2.3.2**: It should be noted that the size of training data is considered as small, and the data ratio of convective and stratiform is not well balanced. Much more data from various precipitation events should be included in the training process if the proposed algorithm is implemented in operation.
**In section conclusion**: Moreover, only very limited training data is used in the current work. Much more data from various precipitation events should be included in the training process if the proposed algorithm is implemented in operation.

*Finally, I retain answers to Reviewers acceptable, but not fully addressed in the text. In other words, answers give a detailed justification to the points indicated by Reviews but the revised text in most cases addresses the point in a sentence not sufficiently complete. My general comment is to take answers to Reviewers and insert them in the text as much as possible "as is". In particular, I retain necessary the justifications for the following points:*
*1. The selection of the datasets, as previously indicated;*

**Response**: Please refer to the response to the first comment.

The explanation for the 11 mm/hour value, citing the two formulas and the threshold of 40 dBZ with its reference;
The sentences on the QPE accuracy have to be revised according to the answer to Reviewer (in my opinion much clear than the sentence in the text), and the paper of Kirsch et al. 2019 have to be included in references;

**Response**: Following the reviewer's comment, the manuscript is modified as:

Convective type precipitation also produces a higher rainfall rate (R) than stratiform type (Anagnostou, 2004). Given the fact that the radar reflectivity (Z) from a stratiform precipitation generally is less than 40 dBZ (Steiner et al., 1995) (hereafter SHY95), the R estimated from stratiform precipitation is less than 11 mm $hr^{-1}$ following the standard Marshall-Palmer relationship (Z = $200R^{1.6}$). In order to obtain accurate rainfall estimation, different R(Z) relationships according to the precipitation types should be applied in the quantitative precipitation estimation (QPE) (Kirsch et al., 2019).

**Response**: Following the reviewer's comment, following discussions were added into the manuscript as:

It should be noted that the $\rho_{HV}$ threshold of 0.9 is used in the training data selection for both convective and stratiform precipitations. As reported in Park et al. (2009), the liquid phase precipitations (e.g., light to heavy rain) are associated with relatively high $\rho_{HV}$ (>0.92). Other types of precipitations, such as the mixture of rain and hail, wet snow, and crystals may produce low $\rho_{HV}$ (<0.85). Since the $i$ is derived based on the raindrop size distribution assumption, the proposed SVM approach is only valid for liquid phase precipitation classification. The classification of other types of precipitation is not in the scope of this work. Therefore, 0.9 is a reasonable $\rho_{HV}$ threshold in the training data selection. The training data used in this work is from pure liquid precipitation events, and the average $\rho_{HV}$ is above 0.98. Similar training results are expected if higher $\rho_{HV}$ threshold is used in the training data selection.

The sentence "In this work, we only use bright band […] to remove bright band signature" of the answer, has to be addressed better in the text;

**Response**: to address the reviewer's comment, we made the following modification in the revised manuscript.

1.) Move the description about the bright band into section 2.1.
2.) The sentences about bright band was modified as:
   Second, a VPR correction is generally needed on the reflectivity field to reduce the measurement biases because of the melting layer (Zhang et al., 2011). The enhanced backscattering amplitudes of melting hydrometeors within the melting layer (bright band) significantly enhance radar reflectivity. The bright band feature is one of the obvious indicators of stratiform precipitation, and normally can be observed from relatively high EAs (such as above 9.9°). Given the fact that data from 1.4° elevation angle is used within the maximum range of 150 km, and the melting layer is usually around 5 km in Taiwan, the radar data used in this work is well below the melting layer. Therefore, no VPR corrections are applied on the fields of Z and $Z_{DR}$.
3.) The section in training data selection was modified as:
   For example, training data from convective precipitation is generally associated with relatively large reflectivity and high vertically integrated liquid (VIL). On the other hand, stratiform precipitations are generally associated with a prominent bright band signature.

When presenting the two principal study cases, it is necessary explicitly declare that small circle areas will be used for further analysis, reporting the justification of the answer to the Reviewer;

**Response**: following illustration is added into the revised manuscript:

A circle is inserted in Figures 5 and 6 to emphasize a region where BAL and SVM show different performances. Within this circle, $BAL^0$ ($BAL^{-0.5}$) classifies the least (most) echoes as convective, and SVM shows the most similar results as MRMS. The averages of Z and $Z_{DR}$ within this region both show relatively large values (Z > 36 dBZ and $Z_{DR}$ > 0.75 dB) as shown in Figure 6. This is a clear indication of convective type precipitation. Both SVM and $BAL^{-0.5}$ classify most of the area within the red circle as convective, and this result is consistent with the MRMS result. Since the separation indexes within the black circle are below or slightly higher than 0, most of the area is classified as stratiform type by $BAL^0$. For this moment, threshold −0.5 shows better performance than 0. Similar reasons may be applied to other regions.

Even if obvious, the values of the statistical parameters of the pure stratiform case have to be reported for symmetry with the other two cases, as declared in the answer but not done;

**Response**: The following sentence was added in the revision:

These three approaches showed consistent classification results with the MRMS result during an 8-hour period evaluation. For all three approaches, the scores of POD, FAR, CSI, and $R^{CS}$ are 1, 0, 1, and 0, respectively.

Finally, also the justification respect the Zdr calibration bias have to be included in the text.

**Response**: The following justification respect to ZDR calibration bias was added in the text.

[revised manuscript text omitted]